# MolAlign3D: Enhancing Fixed-Dimensional E(3)-Equivariant Latent Space for High-Fidelity 3D Molecular Reconstruction and Editing

Zitao Chen [1 2]   Jiatong Ji [2]   Yinjun Jia [1]   Wei-Ying Ma [1]   Yanyan Lan [1 3]

## Abstract

Recent advances in 3D molecular modeling have achieved high-fidelity structural synthesis, yet these models often lack an explicit and manipulable representation space. To address this, MolFLAE introduced a fixed-dimensional, E(3)-equivariant latent space, providing a novel framework for molecular editing independent of atom counts. However, because its latent space was primarily optimized for geometric reconstruction, it remains semantically shallow and inadequate for comprehensive representation learning. In this work, we propose **MolAlign3D**, which evolves this architecture into a unified semantic-generative engine. By anchoring MolFLAE's manipulable latents with embeddings from a pretrained molecular encoder, we yield a manifold that is both semantically dense and geometrically precise. Experiments show that MolAlign3D achieves high-fidelity molecular reconstruction and attains comparable performance on molecular property prediction benchmarks. Notably, the integration of rich semantic priors significantly enhances zero-shot molecular manipulation, including atom-number editing and latent-space interpolation, outperforming prior fixed-dimensional equivariant latent baseline.

## 1. Introduction

Artificial intelligence has become a transformative tool in drug discovery, where the ability to computationally model 3D molecular structures is essential for understanding and designing novel therapeutics. Within this field, two fundamental paradigms have emerged: 3D Molecular representation learning (MRL) and 3D Molecular Generation. Molecular representation learning focuses on encoding complex geometric and chemical information into informative latent descriptors to support tasks such as property prediction (Zhou et al., 2023; Feng et al., 2023). Conversely, 3D molecular generation seeks to model the underlying distribution of the chemical space to synthesize novel, valid structures(Hoogeboom et al., 2022; Song et al., 2024). Recent years have seen substantial progress in both areas, particularly driven by advances in geometric deep learning and E(3)-equivariant architectures.

Despite their shared goal of advancing molecular understanding and design, these two paradigms exhibit complementary strengths and limitations. On the one hand, recent progress in 3D molecular generation has been largely driven by diffusion-based models that achieve impressive geometric fidelity by iteratively denoising atomic coordinates (Hoogeboom et al., 2022; Song et al., 2023; Xu et al., 2023; Song et al., 2024). However, these models operate directly in the space of variable-sized molecular graphs and do not yield an explicit, fixed-dimensional molecular representation. Consequently, their intermediate features are tightly coupled to the number of atoms and diffusion timesteps, making them difficult to reuse as stable descriptors for downstream tasks or to manipulate through structured latent operations.

On the other hand, 3D molecular representation learning methods aim to learn semantically rich descriptors by encoding molecular geometry and chemical context using graph neural networks or equivariant transformers. Large-scale pre-trained models such as Frad(Feng et al., 2023) and Uni-Mol(Zhou et al., 2023) have shown strong performance on a wide range of molecular property prediction benchmarks. Nevertheless, these representations are not designed to be generative. In particular, models like Uni-Mol employ E(3)-invariant global descriptors (e.g., CLS tokens) that cannot uniquely encode or faithfully reconstruct 3D geometric conformations. Consequently, they do not provide a latent space from which edited or interpolated representations can be reliably decoded back into valid 3D structures, limiting their applicability to molecular design and manipulation tasks.

[1]Institute for AI Industry Research (AIR), Tsinghua University [2]Department of Computer Science and Technology, Tsinghua University [3]Beijing Frontier Research Center for Biological Structure, Tsinghua University. Correspondence to: Yanyan Lan <lanyanyan@air.tsinghua.edu.cn>.

*Proceedings of the 43rd International Conference on Machine Learning*, Seoul, South Korea. PMLR 306, 2026. Copyright 2026 by the author(s).

To address the lack of a stable, manipulable representation in generative modeling, MolFLAE (Chen et al., 2025b) introduced a paradigm shift by mapping variable-sized molecular structures into a unique fixed-dimensional, E(3)-equivariant latent space. By utilizing a constant number of latent nodes regardless of atom counts, MolFLAE enabled a novel mechanism for molecular editing through direct latent operations. While this architecture provided a powerful framework for structural manipulation, its latent manifold was primarily optimized for geometric reconstruction and thus remained semantically shallow. Without the integration of rich semantic priors, MolFLAE is inadequate for comprehensive representation learning or property-guided discovery.

A similar tension between generative fidelity and semantic depth has recently been addressed in computer vision. Breakthroughs such as Representation Autoencoders (RAE) (Zheng et al., 2025) and SVG (Shi et al., 2025) have shifted the paradigm by replacing or anchoring generative latents with powerful self-supervised features (e.g., DINOv2 (Oquab et al., 2024)). These works demonstrate that leveraging semantically-dense, pre-trained representations not only enhances generative stability but also enables the latent space to support both perception and production tasks.

Drawing on these insights, we propose **MolAlign3D**, which combines the fixed-dimensional, manipulable latent design of MolFLAE with high-level semantics from pretrained molecular encoders. By anchoring virtual-node latents to global semantic embeddings, MolAlign3D constructs a unified semantic-geometric latent space that augments equivariant manifolds with pretrained semantic structure. This design preserves the benefits of fixed-dimensional latent editing, while the incorporation of semantic priors leads to substantially improved reconstruction fidelity and more informative representations for downstream property prediction compared to MolFLAE.

Our contributions are summarized as follows:

- We propose **MolAlign3D**, a fixed-dimensional AutoEncoder with a E(3)-equivariant latent manifold **that is both semantically dense and geometrically precise**.

- We show that MolAlign3D achieves **substantial gains in 3D reconstruction fidelity** over MolFLAE, and explore latent diffusion within this anchored latent space.

- Experiments further demonstrate that the anchored latent space of MolAlign3D yields **consistent gains in property prediction and zero-shot molecular manipulation**, addressing the semantic limitations of reconstruction-oriented latent models.

## 2. Related Works

### 2.1. 3D Molecular Generation

3D molecular generation has seen rapid progress driven by advances in deep generative modeling. Early autoregressive approaches, such as G-SchNet(Gebauer et al., 2019) and G-SphereNet (Luo & Ji, 2022), construct molecules via sequential atom attachment, but require careful design of action spaces and generation orderings, which limits their general applicability. More recently, E(3)-equivariant diffusion-like models, such as EDM(Hoogeboom et al., 2022), EquiFM(Song et al., 2023), EDM-Bridge(Wu et al., 2022) and GeoBFN(Song et al., 2024) have emerged as a dominant paradigm by directly modeling distributions over atomic coordinates while preserving E(3) equivariance, achieving strong geometric fidelity and stability on standard benchmarks.

Despite their success in generation quality, these models do not expose a reusable molecular representation. The intermediate features are tightly coupled to diffusion timesteps and noise levels, making them unsuitable as stable molecular descriptors. While latent diffusion models such as GeoLDM (Xu et al., 2023) alleviate this, their latent spaces still scale with molecular size. To overcome this, MolFLAE (Chen et al., 2025b) maps molecules to a fixed-dimensional, E(3)-equivariant manifold, enabling 3D generation via sampling and structural editing via latent arithmetic. However, its reconstruction-oriented training yields a semantically shallow space inadequate for property-aware tasks.

### 2.2. Molecular Representation Learning

Molecular representation learning (MRL) aims to learn transferable and semantically meaningful molecular embeddings that can generalize across diverse downstream task. Early approaches primarily focused on 1D representations such as SMILES strings(Xu et al., 2017; Honda et al., 2019; Wang et al., 2019; Chithrananda et al., 2020) using sequence models, or 2D graphs employing self-supervised objectives such as masked prediction(Rong et al., 2020; ZHANG et al., 2021; Fang et al., 2022) or 2D-3D contrastive learning(Li et al., 2022; Wang et al., 2022). Modern 3D-based models achieve state-of-the-art results by leveraging geometric tasks like coordinate denoising(Jiao et al., 2023; Zhou et al., 2023; Liu et al., 2023) and force field learning (Feng et al., 2023) to capture fine-grained spatial relationships, which has proven critical for downstream physical and chemical tasks.

Despite their success, most existing MRL representations remain primarily discriminative and lack the ability to support faithful 3D reconstruction. While recent frameworks such as UniGEM (Feng et al., 2025) begin to unify property prediction with molecular generation, their instance-dependent

designs limit latent manipulation. The broader difficulty is that strong semantic encoders and generative architectures have developed largely independently. Methods that excel at representation learning are not designed to produce reconstructible, editable latent spaces, and vice versa.

## 2.3. Molecular Editing

Molecular editing encompasses a broad range of structural transformations like linker design (Huang et al., 2022; Igashov et al., 2024) and scaffold hopping (Hu et al., 2023). Traditional 3D methods often treat these as conditional generation tasks. For instance, LinkerNet (Guan et al., 2023) and Delete (Chen et al., 2025a) utilize explicit conditioning via structural masks or predefined fragments. While effective, these approaches are inherently limited by specific problem formulations, requiring distinct supervised signals or rigid constraints for each editing scenario.

To improve flexibility, MolFLAE (Chen et al., 2025b) reformulated editing as continuous latent space traversal, enabling zero-shot manipulation via vector arithmetic. However, its reconstruction-centric manifold lacks the semantic linearity required for smooth structure-property co-interpolation and the structural disentanglement needed for complex edits like cross-component swapping.

## 2.4. Bridging Generative and Global Semantic Representations

The prevailing paradigm in visual generation, established by LDM (Rombach et al., 2022), typically relies on a VAE to compress images into a latent space for diffusion. However, its reconstruction-centric latent space often lacks clear semantic separation and discriminative structure(Xu et al., 2025; Lee et al., 2025), limiting its utility as a unified representation for both generation and perception.

To bridge this gap, recent studies have leveraged pretrained vision models like DINOv2(Oquab et al., 2024) for representation alignment, providing regularization for the latent manifold or accelerating training convergence (Yao et al., 2025; Yu et al., 2025). Beyond alignment, RAE(Zheng et al., 2025) replaces traditional VAEs with pretrained foundation models to prioritize high-level semantics as the generative substrate. Concurrently, SVG (Shi et al., 2025) employs a decoupled latent space, utilizing self-supervised anchors for global consistency and a residual branch for fine-grained reconstruction.

These advances suggest a promising direction for resolving the tension between generative fidelity and semantic depth, yet transferring these ideas to molecular modeling raises distinct challenges. Molecular structures require permutation invariance and E(3)-equivariance, which standard vision architectures lack. Molecules couple discrete chemi-

cal identities with continuous 3D geometry, demanding joint modeling of combinatorial constraints and precise spatial arrangements. Molecular modeling must also account for multiple conformations of the same molecule that differ in 3D geometry but share identical properties. MolAlign3D addresses these gaps by building upon a fixed-dimensional E(3)-equivariant latent space, decoupling the representation into coordinate and feature branches, and enforcing multi-conformer contrastive consistency.

## 3. Methodology

**Overview.** Our framework consists of two parts: (i) **MolAlign3D** (Sec. 3.1), an autoencoder that encodes molecules into a fixed-dimensional E(3)-equivariant latent space, anchors the latent features with pretrained semantics, and decodes structures via a Bayesian Flow Network; and (ii) a **Latent Diffusion Model** (Sec. 3.2) that learns to generate the anchored latent representations of MolAlign3D for unconditional molecular generation. The overall architecture is illustrated in Fig. 1.

### 3.1. MolAlign3D

We now detail the autoencoder, which comprises three modules: an E(3)-equivariant encoder, a pretrained semantic anchoring mechanism, and a Bayesian Flow Network decoder.

**Encoder.** The encoder is responsible for transforming the input molecule $\mathcal{M}$ into a fixed-dimensional latent representation. The molecule is represented as $\mathcal{M} = (\mathbf{X}_{\mathcal{M}}, \mathbf{V}_{\mathcal{M}})$, where $\mathbf{X}_{\mathcal{M}} \in \mathbb{R}^{N_{\mathcal{M}} \times 3}$ contains the 3D Cartesian coordinates of the atoms, and $\mathbf{V}_{\mathcal{M}} \in \{0, 1\}^{N_{\mathcal{M}} \times K}$ represents the atomic species using one-hot vectors over $K$ atom types. After centering the molecule by translating its centroid to the origin, the atomic one-hot vectors are mapped to continuous node features via a learnable embedding layer, yielding a node feature matrix $\mathbf{H}_{\mathcal{M}} \in \mathbb{R}^{N_{\mathcal{M}} \times D_Z}$, where $D_Z$ denotes the latent feature dimensionality.

In addition to the atomic nodes, we introduce $N_Z$ virtual nodes, denoted as $\mathcal{Z} = (\mathbf{X}_{\mathcal{Z}}, \mathbf{H}_{\mathcal{Z}})$, where the virtual-node coordinates are initialized as $\mathbf{X}_{\mathcal{Z}} = \mathbf{0} \in \mathbb{R}^{N_Z \times 3}$, and $\mathbf{H}_{\mathcal{Z}} \in \mathbb{R}^{N_Z \times D_Z}$ are learnable feature embeddings. These atomic and virtual nodes are then concatenated and processed jointly by an E(3)-equivariant backbone $\mathbf{\Phi}_{\theta}^{\mathrm{E}(3)}$:

$$\big(\widetilde{\mathbf{X}}_{\mathcal{M}}, \widetilde{\mathbf{H}}_{\mathcal{M}}, \widetilde{\mathbf{X}}_{\mathcal{Z}}, \widetilde{\mathbf{H}}_{\mathcal{Z}}\big) = \mathbf{\Phi}_{\theta}^{\mathrm{E}(3)}\Big(\big[\mathbf{X}_{\mathcal{M}} \oplus \mathbf{X}_{\mathcal{Z}}\big], \\ \big[\mathbf{H}_{\mathcal{M}} \oplus \mathbf{H}_{\mathcal{Z}}\big]\Big), \quad (1)$$

where $\oplus$ denotes concatenation along the node dimension.

From the output of the backbone, we retain only the virtual-node results, which are denoted as equivariant coordinates

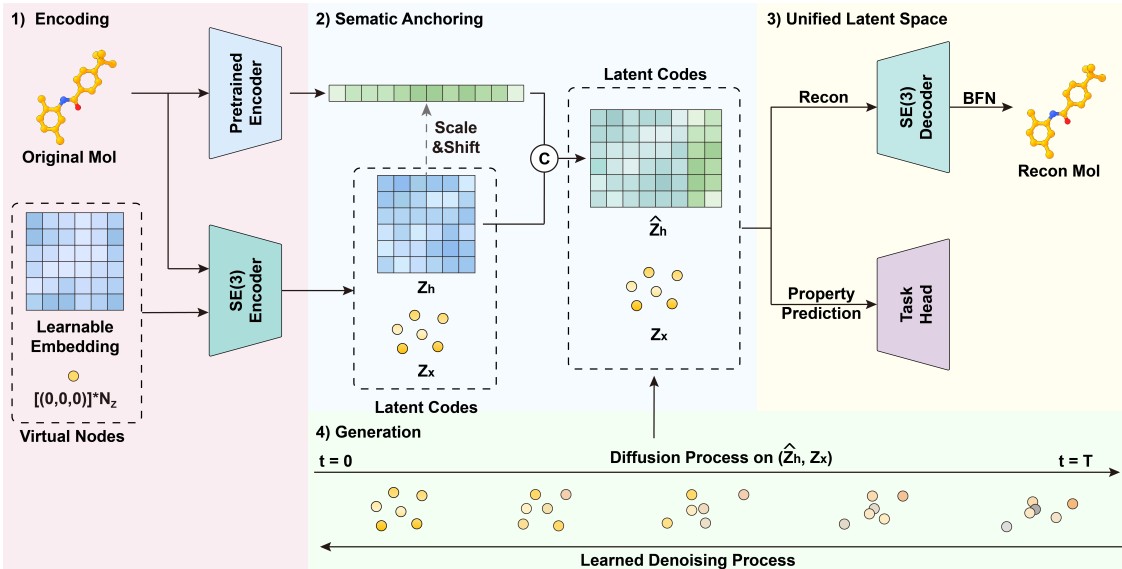

*Figure 1.* Overview of MolAlign3D and LDM model. MolAlign3D encodes an input molecule into a fixed-dimensional latent space using an E(3)-equivariant encoder, producing coordinate latents $\mathbf{Z}_x$ and feature latents $\mathbf{Z}_h$. The feature latents are semantically anchored by a frozen pretrained molecular encoder to obtain $\widehat{\mathbf{Z}}_h$, forming a unified latent representation $\boldsymbol{\zeta} = [\mathbf{Z}_x; \widehat{\mathbf{Z}}_h]$. A Bayesian Flow Network decodes $\boldsymbol{\zeta}$ into discrete atom types and 3D geometries, while a latent diffusion model enables molecular generation directly in the latent space. The learned latent representation also supports downstream property prediction via a lightweight task head.

and invariant features:

$$\left( \boldsymbol{\mu}_x, \mathbf{Z}_h \right) \coloneqq \left( \widetilde{\mathbf{X}}_{\mathcal{Z}}, \widetilde{\mathbf{H}}_{\mathcal{Z}} \right) \in \mathbb{R}^{N_Z \times (3 + D_Z)}. \qquad (2)$$

Here $\boldsymbol{\mu}_x \in \mathbb{R}^{N_Z \times 3}$ denotes the equivariant coordinate output of the encoder for the virtual nodes, and $\mathbf{Z}_h \in \mathbb{R}^{N_Z \times D_Z}$ denotes the corresponding invariant feature embeddings. By construction, rigid transformations of the centered input molecule induce corresponding transformations in the latent coordinates, which ensures E(3)-equivariance. A formal proof of this property is provided in Appendix A.

To encourage smooth and stable geometric variations in the latent space, we adopt an asymmetric variational parameterization only on the coordinate latent:

$$\mathbf{Z}_x = \boldsymbol{\mu}_x + \boldsymbol{\varsigma}_x \odot \boldsymbol{\varepsilon}, \qquad \boldsymbol{\varepsilon} \sim \mathcal{N}(\mathbf{0}, \mathbf{I}), \qquad (3)$$

where the variance $\boldsymbol{\varsigma}_x^2$ is predicted through latent states that produce $\mathbf{Z}_h$. Also, to stabilize training and improve semantic coherence, we introduce auxiliary regularization terms, with their full formulation provided in Appendix C. The final fixed-dimensional latent representation produced by the encoder is a concatenation of the coordinate and feature latents:

$$\boldsymbol{z} = [\mathbf{Z}_x; \mathbf{Z}_h]. \qquad (4)$$

**Pretrained Semantic Anchoring.** To enrich the learned latent features with global semantic information, we leverage a frozen pretrained molecular encoder $\mathfrak{f}_{\mathrm{pre}}(\cdot)$, which generates a global invariant embedding for the molecule $\mathcal{M}$:

$$\mathbf{u} = \mathfrak{f}_{\mathrm{pre}}(\mathcal{M}) \in \mathbb{R}^{D_u}. \qquad (5)$$

To align the scale of the feature latent $\mathbf{Z}_h$ with the pretrained embedding $\mathbf{u}$, we perform affine moment matching. Denote the empirical mean and standard deviation of $\mathbf{Z}_h$ as $\overline{Z}_h \in \mathbb{R}$ and $\varsigma_{Z_h} \in \mathbb{R}$, respectively. Similarly, we define the mean $\overline{u} \in \mathbb{R}$ and standard deviation $\varsigma_u \in \mathbb{R}$ for $\mathbf{u}$. The moment-aligned features are then computed as:

$$\widetilde{\mathbf{Z}}_h = \frac{\mathbf{Z}_h - \mathbf{1}\,\overline{Z}_h}{\varsigma_{Z_h} + \epsilon}\, \varsigma_u + \mathbf{1}\,\overline{u}, \qquad (6)$$

where $\mathbf{1}$ is an all-ones column vector of length $N_Z$.

Next, we partition the molecule-level embedding $\mathbf{u}$ into $N_Z$ node-level segments:

$$\mathbf{U}_{\mathrm{split}} = \left[ \mathbf{u}^{(1)}; \mathbf{u}^{(2)}; \ldots; \mathbf{u}^{(N_Z)} \right] \in \mathbb{R}^{N_Z \times D_u'}, \qquad (7)$$

where $D_u' = D_u / N_Z$. These per-node semantics are then concatenated with the moment-aligned features:

$$\widehat{\mathbf{Z}}_h = \mathrm{Concat}\left( \widetilde{\mathbf{Z}}_h, \ \mathbf{U}_{\mathrm{split}} \right) \in \mathbb{R}^{N_Z \times (D_Z + D_u')}. \qquad (8)$$

This process enriches each virtual node $j$ with a local feature vector $\widehat{\mathbf{Z}}_h^{(j)}$ that carries global semantics, while maintaining the E(3)-equivariance of the coordinate branch (see Appendix B). Overall, the complete molecular representation is given by the anchored latent

$$\boldsymbol{\zeta} = [\mathbf{Z}_x; \widehat{\mathbf{Z}}_h], \qquad (9)$$

which provides a fixed-dimensional, E(3)-equivariant geometric–semantic encoding of the molecule.

**Decoder.** Given the anchored latent space constructed by the encoder and semantic alignment modules, the decoder is responsible for mapping latent representations back to discrete atom types and continuous 3D molecular geometries.

To encourage the decoder to generalize to mildly off-manifold latent inputs, we apply stochastic smoothing to the feature branch following (Zheng et al., 2025):

$$\boldsymbol{\zeta}' = [\mathbf{Z}_x; \widehat{\mathbf{Z}}_h + \mathbf{n}], \quad \mathbf{n} \sim \mathcal{N}(\mathbf{0}, \sigma^2 \mathbf{I}), \quad \sigma \sim \mathcal{HN}(\tau), \tag{10}$$

where $\mathcal{HN}(\tau)$ denotes a half-normal distribution with scale $\tau$. Noise is injected only into the semantic feature latent $\widehat{\mathbf{Z}}_h$ for the reason that uncertainty in the coordinate branch $\mathbf{Z}_x$ is already introduced through the asymmetric reparameterization in the encoder. Importantly, the noise is used solely during the training process of decoder for robustness enhancement, while the underlying molecular representation remains a clean, deterministic latent embedding.

The Bayesian Flow Network (BFN) decoder models the conditional distribution $p(\mathbf{X}_{\mathcal{M}}, \mathbf{V}_{\mathcal{M}} \mid \boldsymbol{\zeta}')$ by maintaining a belief state $\boldsymbol{\Theta}_i = (\boldsymbol{\mu}_i^x, \boldsymbol{\rho}_i^x, \boldsymbol{\theta}_i^v)$ at each communication step $i \in \{1, \ldots, T\}$. The coordinates are parameterized by $(\boldsymbol{\mu}_i^x, \boldsymbol{\rho}_i^x)$, representing factorized Gaussians, while $\boldsymbol{\theta}_i^v$ parameterizes categorical beliefs for atom types.

To generate noisy observations of the molecule, the sender perturbs the ground-truth coordinates and atom types. For coordinates:

$$p_S\big(\mathbf{Y}_x^{(i)} \mid \mathbf{X}_{\mathcal{M}}; \alpha_i\big) = \mathcal{N}\big(\mathbf{Y}_x^{(i)} \mid \mathbf{X}_{\mathcal{M}}, \alpha_i^{-1}\mathbf{I}\big), \tag{11}$$

where $\alpha_i$ is a precision schedule. For discrete atom types, we use the Gaussian-embedding trick:

$$p_S\big(\mathbf{Y}_v^{(i)} \mid \mathbf{V}_{\mathcal{M}}; \alpha_i'\big) = \mathcal{N}\big(\mathbf{Y}_v^{(i)} \mid \alpha_i'(K\mathbf{V}_{\mathcal{M}} - \mathbf{1}), \alpha_i'K\mathbf{I}\big). \tag{12}$$

Given the noisy observations $\mathbf{Y}^{(i)} = (\mathbf{Y}_x^{(i)}, \mathbf{Y}_v^{(i)})$, the receiver outputs a reconstruction $\widehat{\mathcal{M}} = (\widehat{\mathbf{X}}, \widehat{\mathbf{V}})$ via:

$$p_O\big(\widehat{\mathcal{M}} \mid \mathbf{Y}^{(i)}; \boldsymbol{\Theta}_{i-1}, \boldsymbol{\zeta}'\big) = \Phi\big(\boldsymbol{\Theta}_{i-1}, \boldsymbol{\zeta}', t_i\big), \tag{13}$$

where $\Phi$ is a neural network that consumes the belief state, the perturbed anchored latent, and the current step $t_i$. Reapplying sender noise to the reconstructed molecule $\widehat{\mathcal{M}}$ yields the receiver-implied observation distribution:

$$p_R\big(\mathbf{Y}^{(i)} \mid \boldsymbol{\Theta}_{i-1}, \boldsymbol{\zeta}'; t_i\big) = \\ \mathbb{E}_{\widehat{\mathcal{M}} \sim p_O}\big[p_S\big(\mathbf{Y}^{(i)} \mid \widehat{\mathcal{M}}; \alpha_i, \alpha_i'\big)\big]. \tag{14}$$

The training process minimizes the per-step KL divergence $\mathrm{KL}\big(p_S(\cdot \mid \mathcal{M}; \alpha_i, \alpha_i') \,\|\, p_R(\cdot \mid \boldsymbol{\Theta}_{i-1}, \boldsymbol{\zeta}'; t_i)\big)$. Since both the sender and receiver coordinate components are Gaussian

with the same covariance, the coordinate KL reduces to a closed-form squared error:

$$\mathcal{L}_x^{(i)} = D_{\mathrm{KL}}\Big(\mathcal{N}(\mathbf{X}_{\mathcal{M}}, \alpha_i^{-1}\mathbf{I}) \,\Big\|\, \mathcal{N}(\widehat{\mathbf{X}}(\boldsymbol{\Theta}_{i-1}, \boldsymbol{\zeta}', t_i), \alpha_i^{-1}\mathbf{I})\Big)$$
$$= \frac{\alpha_i}{2}\big\|\mathbf{X}_{\mathcal{M}} - \widehat{\mathbf{X}}(\boldsymbol{\Theta}_{i-1}, \boldsymbol{\zeta}', t_i)\big\|_2^2. \tag{15}$$

For atom types, the loss can also be derived by taking KL-divergence between Gaussians(Graves et al., 2023):

$$\mathcal{L}_v^{(i)} = \ln \mathcal{N}\Big(\mathbf{Y}_v^{(i)} \,\Big|\, \alpha_i'(K\mathbf{e}_{v_M} - \mathbf{1}), \alpha_i'K\mathbf{I}\Big)$$
$$- \sum_{d=1}^{N_{\mathcal{M}}} \ln\Bigg( \sum_{k=1}^{K} p_O\big(k \mid \boldsymbol{\Theta}_{i-1}, \boldsymbol{\zeta}', t_i\big) \times$$
$$\mathcal{N}\Big(\cdot^{(d)} \,\Big|\, \alpha_i'(K\mathbf{e}_k - \mathbf{1}), \alpha_i'K\mathbf{I}\Big)\Bigg). \tag{16}$$

Combining both terms, the $n$-step objective is:

$$\mathcal{L}_n(\mathcal{M}, \boldsymbol{\zeta}') = \mathbb{E}_{i \sim \mathcal{U}(1,n)}\Big[\mathcal{L}_x^{(i)} + \mathcal{L}_v^{(i)}\Big]. \tag{17}$$

The belief-state updates are given in Appendix E.

### 3.2. Latent Diffusion Model

Since the anchored semantic latent $\widehat{\mathbf{Z}}_h$ does not follow a simple factorized prior, we model generation via a joint latent diffusion process defined directly on the anchored latent space $\boldsymbol{\zeta} = [\mathbf{Z}_x; \widehat{\mathbf{Z}}_h]$. Concretely, we adopt a cosine variance schedule for the forward noise levels $\{\alpha_t\}_{t=1}^T$ and perform reverse sampling via standard DDPM ancestral transitions with $T = 100$ steps, following the setup of UniMoMo (Kong et al., 2025). Let $\mathbf{Z}_0^{\{\cdot\}} \in \{\mathbf{Z}_x, \widehat{\mathbf{Z}}_h\}$ denote the clean encoder latents. For a discrete timestep $t \in \{1, \ldots, T\}$, the forward noising process is defined as

$$q(\mathbf{Z}_t^{\{\cdot\}} \mid \mathbf{Z}_0^{\{\cdot\}}) = \mathcal{N}\Big(\mathbf{Z}_t^{\{\cdot\}} \mid \sqrt{\bar{\alpha}_t}\,\mathbf{Z}_0^{\{\cdot\}}, (1 - \bar{\alpha}_t)\mathbf{I}\Big), \tag{18}$$

which admits the reparameterization

$$\mathbf{Z}_t^{\{\cdot\}} = \sqrt{\bar{\alpha}_t}\,\mathbf{Z}_0^{\{\cdot\}} + \sqrt{1 - \bar{\alpha}_t}\,\boldsymbol{\varepsilon}^{\{\cdot\}}, \quad \boldsymbol{\varepsilon}^{\{\cdot\}} \sim \mathcal{N}(\mathbf{0}, \mathbf{I}), \tag{19}$$

where $\bar{\alpha}_t = \prod_{s=1}^t \alpha_s$ and $\{\cdot\} \in \{x, h\}$ indexes the coordinate and semantic branches.

To rescale the noise level and improves diffusion behavior, we adopt a continuous-time schedule shift following (Esser et al., 2024; Zheng et al., 2025). Defining $\tau = t/T$, the effective diffusion time is

$$\tilde{\tau} = \frac{\alpha\,\tau}{1 + (\alpha - 1)\tau}, \quad \alpha = \sqrt{\frac{m}{n_{\text{base}}}}, \tag{20}$$

where $m$ denotes the per-sample latent dimensionality, $n_{base} = 128$. The shifted time $\tilde{\tau}$ is discretized to obtain an effective diffusion index $\tilde{t}$. The reverse process is

parameterized by an E(3)-equivariant denoising network $\mathcal{F}_\psi(\mathbf{Z}_t^x, \mathbf{Z}_t^h, \tilde{t})$, which directly predicts the clean latents $\widehat{\mathbf{Z}}_0^{\{\cdot\}}$ from noisy inputs. Under the $x_0$-prediction parameterization, the reverse-time transition is given by

$$p_\psi(\mathbf{Z}_{t-1}^{\{\cdot\}} \mid \mathbf{Z}_t^{\{\cdot\}}) = \mathcal{N}\Big(\mathbf{Z}_{t-1}^{\{\cdot\}} \mid \boldsymbol{\mu}_\psi^{\{\cdot\}}(\mathbf{Z}_t^{\{\cdot\}}, \tilde{t}), \beta_t\mathbf{I}\Big), \quad (21)$$

where $\beta_t = 1 - \alpha_t$ controls the variance of the reverse transition, with the mean computed from the predicted clean latent as

$$\boldsymbol{\mu}_\psi^{\{\cdot\}} = \frac{1}{\sqrt{\alpha_t}}\left(\mathbf{Z}_t^{\{\cdot\}} - \frac{\beta_t}{\sqrt{1-\bar{\alpha}_t}}\left(\mathbf{Z}_t^{\{\cdot\}} - \sqrt{\bar{\alpha}_t}\,\widehat{\mathbf{Z}}_0^{\{\cdot\}}\right)\right). \tag{22}$$

Training minimizes the mean-squared error between the predicted clean latents and the ground-truth encoder outputs, averaged over all virtual nodes:

$$\mathcal{L}_{\text{LDM}} = \mathbb{E}_{t,\boldsymbol{\varepsilon}}\left[\sum_{\{\cdot\}\in\{x,h\}}\frac{1}{N_Z}\sum_{j=1}^{N_Z}\|\widehat{\mathbf{Z}}_0^{\{\cdot\},(j)} - \mathbf{Z}_0^{\{\cdot\},(j)}\|_2^2\right]. \tag{23}$$

## 4. Experiments

In this section, we evaluate the quality of the fixed-length latent space of MolAlign3D, focusing on reconstruction fidelity, *diffusability* (Skorokhodov et al., 2025), and the preservation of pretrained semantic representations. We further analyze the smoothness and disentanglement of the latent manifold in zero-shot manipulation tasks. Throughout all experiments, we adopt Uni-Mol(Zhou et al., 2023) as the pretrained molecular encoder and use the same latent dimensionality and training configurations across all models; detailed settings are provided in Appendix G.

### 4.1. Reconstruction and Unconditional Generation

We begin by evaluating whether anchoring pretrained semantic representations improves reconstruction of 3D molecular geometries while remaining compatible with diffusion-based unconditional generation. Unlike MolFLAE, which admits direct sampling from an explicit latent prior, MolAlign3D relies on latent diffusion to model and generate anchored representations. Importantly, since high reconstruction accuracy alone does not guarantee a well-structured, diffusable latent space (Yao et al., 2025), we jointly assess reconstruction fidelity and unconditional generation quality.

**Setup.** We conduct experiments on GEOM-Drugs (Axelrod & Gómez-Bombarelli, 2020), which contains approximately 430K drug-like molecules with up to 181 atoms. Following EDM (Hoogeboom et al., 2022), we retain up to 30 lowest-energy conformers per molecule, yielding approximately 6.9 million conformers. Compared to smaller benchmarks

such as QM9 (Ramakrishnan et al., 2014), GEOM-Drugs is substantially more challenging due to its scale and conformational diversity. Reconstruction is evaluated by encoding–decoding test molecules, while unconditional generation is performed by sampling 10,000 latent codes and decoding them into molecular structures. We report Atom Stability, Validity, and drug-likeness metrics (Appendix F).

**Baselines.** For reconstruction, we compare primarily against MolFLAE (Chen et al., 2025b), which shares the same autoencoding architecture but lacks semantic anchoring. We additionally construct a Uni-Mol decoding baseline, where the semantic latent $\widehat{Z}_h$ is fully derived from Uni-Mol features and the coordinate latent $Z_x$ is fixed at the molecular center, consistent with Uni-Mol's treatment of the [CLS] token. For unconditional generation, we further compare against representative 3D molecular generative models, including EDM (Hoogeboom et al., 2022), GeoLDM (Xu et al., 2023), GeoBFN (Song et al., 2024), and UniGEM (Feng et al., 2025).

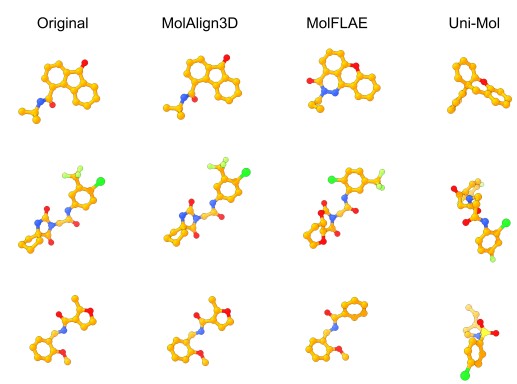

*Figure 2.* Qualitative reconstruction examples on GEOM-Drugs. Columns show the ground-truth conformation (Original) and reconstructions by MolAlign3D, MolFLAE, and the Uni-Mol baseline in a shared coordinate frame.

**Results** Table 1 shows that MolAlign3D substantially outperforms all baselines on both MCS-IoU and Shape Tanimoto(Appendix F), indicating superior preservation of molecular scaffolds and fine-grained 3D geometry. Compared to MolFLAE, MolAlign3D improves MCS-IoU from 46.11% to 80.79% and Shape Tanimoto from 49.84% to 64.91%. It suggests that semantic anchoring provides a strong global prior that regularizes the fixed-length latent space toward chemically meaningful regions while preserving precise conformations, as further illustrated in Fig. 2.

In contrast, the Uni-Mol decoding baseline performs poorly despite strong pretrained semantics, often producing distorted or topologically inconsistent structures. This indicates that semantics alone are insufficient for faithful 3D reconstruction without a geometry-aware latent space.

*Table 1.* Reconstruction performance and latent space characteristics on GEOM-Drugs.

| Model | MCS-IoU (%,↑) | Shape Tanimoto (%, ↑) | Semantic | Reconstruction | |
| --- | --- | --- | --- | --- | --- |
| | | | | Topo | Conformation |
| Uni-Mol | 34.35 | 26.86 | ✓ | ✗ | ✗ |
| MolFLAE | 46.11 | 49.84 | ✗ | ✓ | ✓ |
| **MolAlign3D** | **80.79** | **64.91** | ✓ | ✓ | ✓ |

Beyond reconstruction, we evaluate whether the anchored latent space remains amenable to unconditional generation. Table 2 shows that MolAlign3D achieves strong Atom Stability on GEOM-Drugs while producing drug-likeness statistics that closely match the data distribution (e.g., QED, Lipinski, and LogP). Although Validity and SA are slightly below the best baseline in some cases, MolAlign3D generates molecules that are overall more consistent with the reference set across multiple complementary metrics.

*Table 2.* Unconditional generation results on GEOM-Drugs. We report Atom Stability, Validity, and drug-likeness metrics.

| Model | Atom Sta. (%, ↑) | Valid (%, ↑) | QED (↑) | SA (↑) | Lipinski (↑) | LogP |
| --- | --- | --- | --- | --- | --- | --- |
| *Data* | *86.5* | *99.9* | *0.64* | *0.84* | *4.80* | *2.99* |
| EDM | 81.4 | 96.6 | 0.36 | 0.59 | 4.27 | 0.77 |
| GeoLDM | 84.5 | 99.4 | 0.40 | 0.62 | 4.31 | 0.66 |
| GeoBFN | 86.2 | 91.7 | - | - | - | - |
| UniGEM | 85.0 | 98.4 | 0.36 | 0.63 | 4.24 | 0.61 |
| MolFLAE | 86.6 | **99.7** | 0.60 | **0.75** | 4.75 | 1.38 |
| **MolAlign3D** | **87.2** | 99.2 | **0.64** | 0.69 | **4.79** | 2.96 |

## 4.2. Molecular Property Prediction

Beyond reconstruction and generation, a key limitation of prior latent molecular autoencoders such as MolFLAE is the *semantic shallowness* of their learned representations. Although MolFLAE preserves 3D geometry and supports latent manipulation, its latent space is primarily reconstruction-oriented rather than optimized for capturing high-level molecular semantics, limiting downstream discriminative performance. To examine whether semantic anchoring in MolAlign3D addresses this limitation, we evaluate its latent representations on standard molecular property prediction tasks.

**Setup.** We evaluate molecular representations on the MoleculeNet benchmark (Wu et al., 2017), covering diverse classification and regression tasks. Following Uni-Mol (Zhou et al., 2023), we adopt scaffold splitting with an 8:1:1 train/validation/test ratio and perform hyperparameter selection using the same protocol. Additional evaluation details are provided in Appendix K.

**Baselines.** We compare against Uni-Mol (Zhou et al., 2023) and the baselines reported in its benchmark, including supervised GNN methods (D-MPNN (Yang et al., 2019), AttentiveFP (Xiong et al., 2020)) and pretraining-based approaches such as PretrainGNN (Hu et al., 2020), GraphMV (Liu et al., 2022), MolCLR (Wang et al., 2022), and GEM

(Fang et al., 2022). We additionally include MolFLAE (Chen et al., 2025b) as an internal baseline to isolate the effect of semantic anchoring, as both methods share the same backbone and training pipeline.

**Results.** MolAlign3D achieves performance comparable to Uni-Mol across both classification and regression tasks, with strong ROC-AUC on BBBP, BACE, and ClinTox, and low prediction errors on ESOL, FreeSolv, and Lipophilicity. It remains competitive on SIDER and QM7, and consistently outperforms MolFLAE across all tasks.

*Table 3.* Molecular property prediction on MoleculeNet. Classification results are reported in ROC-AUC (%), and regression results are reported in RMSE for ESOL, FreeSolv, and Lipophilicity, and MAE for QM7.

| Model | Classification | | | | Regression | | | |
| --- | --- | --- | --- | --- | --- | --- | --- | --- |
| | BBBP | BACE | ClinTox | SIDER | ESOL | FreeSolv | Lipo | QM7 |
| D-MPNN | 71.0 | 80.9 | 90.6 | 57.0 | 1.050 | 2.082 | 0.683 | 103.5 |
| Attentive FP | 64.3 | 78.4 | 84.7 | 60.6 | 0.877 | 2.073 | 0.721 | 72.0 |
| PretrainGNN | 68.7 | 84.5 | 72.6 | 62.7 | 1.100 | 2.764 | 0.739 | 113.2 |
| GraphMVP | 72.4 | 81.2 | 79.1 | 63.9 | 1.029 | – | 0.681 | – |
| MolCLR | 72.2 | 82.4 | 91.2 | 58.9 | 1.271 | 2.594 | 0.691 | 66.8 |
| GEM | 72.4 | 85.6 | 90.1 | 67.2 | 0.798 | 1.877 | 0.660 | 58.9 |
| Uni-Mol | **72.9** | **85.7** | 91.9 | 65.9 | 0.788 | 1.480 | 0.603 | **41.8** |
| MolFLAE | 69.6 | 80.7 | 75.2 | 61.5 | 0.926 | 1.862 | 0.835 | 74.3 |
| MolAlign3D | 72.7 | 85.6 | 93.7 | 66.3 | **0.778** | **1.445** | **0.595** | 51.7 |

Overall, these results indicate that semantic anchoring effectively alleviates the semantic shallowness of MolFLAE, yielding more informative representations for downstream prediction without sacrificing geometric fidelity, and positioning MolAlign3D as a unified representation for both generative and discriminative molecular modeling.

## 4.3. Zero-Shot Molecular Manipulation

Having demonstrated improved reconstruction fidelity and more semantically informative representations, we finally examine whether MolAlign3D preserves the latent editability that characterizes MolFLAE across three editing tasks defined in the MolFLAE framework.

**Setup.** Following MolFLAE, we conduct all zero-shot molecular manipulation experiments on ZINC-9M (the in-stock subset of ZINC (Irwin et al., 2020)) dataset. Models are evaluated without any task-specific finetuning, and all editing operations are performed directly in the fixed-dimensional latent space. We use the same data preprocessing, evaluation protocols, and metrics as in MolFLAE to ensure a fair comparison.

### 4.3.1. ATOM NUMBER EDITING

**Task.** This task evaluates the model's ability to generate structural analogs under controlled changes in molecular size. Given a source molecule, we encode it into the latent space and decode it while conditioning on a target atom number $N_{new} = N_{source} \pm k$, where $k \in \{1, 2\}$. Structural

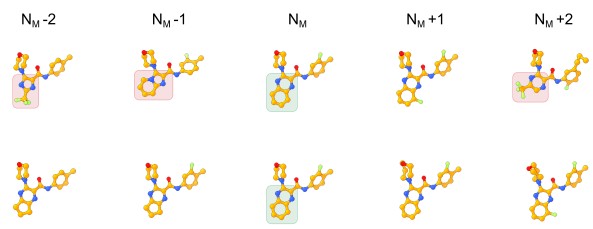

*Figure 3.* Atom number editing. MolFLAE (top) and MolAlign3D (bottom) are compared under atom insertion and deletion. MolAlign3D preserves the central bicyclic scaffold (green boxes) and privileged substructures, absorbing atom-count changes through peripheral modifications. MolFLAE is more sensitive to atom number changes, with larger insertions or deletions disrupting the core backbone (red boxes).

preservation is measured using Maximum Common Substructure Intersection-over-Union (MCS-IoU), along with Shape Tanimoto similarity.

*Table 4.* Editing performance on ZINC. Metrics are reported across different atom number variations.

| Metric | Model | -2 | -1 | 0 | +1 | +2 |
|---|---|---|---|---|---|---|
| MCS-IoU (%,↑) | MolFLAE | 70.00 | 75.95 | 83.80 | 74.28 | 69.76 |
| | **MolAlign3D** | **70.26** | **77.35** | **87.78** | **78.45** | **71.14** |
| Shape Tanimoto (%, ↑) | MolFLAE | 60.12 | 62.65 | 66.77 | 62.57 | 60.26 |
| | **MolAlign3D** | **60.94** | **63.99** | **69.70** | **64.27** | **61.47** |

**Results.** Table 4 summarizes the atom-number editing results, and Figure 3 provides representative examples. Across all settings, MolAlign3D consistently improves both MCS-IoU and Shape Tanimoto over MolFLAE, with the largest gains around small perturbations. This behavior supports the intuition that semantic anchoring yields a more robust, scaffold-level latent representation. When decoding under a mismatched atom-count constraint, edits are more likely to be absorbed as peripheral modifications rather than breaking the core backbone.

### 4.3.2. STRUCTURE DISENTANGLEMENT AND RECONSTRUCTION

**Task.** We evaluate the disentanglement between the spatial latent $Z_x$ and the semantic feature latent $\widehat{Z}_h$ using two hybrid molecules. Given two molecules $A$ and $B$, Preserving $Z_x$ is constructed as $[Z_x^{(A)}, \widehat{Z}_h^{(B)}]$, while Preserving $\widehat{Z}_h$ is constructed as $[Z_x^{(B)}, \widehat{Z}_h^{(A)}]$. Both hybrids are decoded and evaluated against molecule $A$.

**Results.** Table 6 reports hybrid decoding results under controlled preservation of $Z_x$ or $\widehat{Z}_h$. When preserving $Z_x$, MolAlign3D maintains comparable shape similarity while reducing MACCS similarity, indicating stronger geometric control with greater semantic flexibility. Conversely, when preserving $\widehat{Z}_h$, MolAlign3D achieves substantially higher

MACCS similarity at similar geometric deviation, reflecting improved semantic faithfulness. Overall, MolAlign3D exhibits a cleaner functional separation, with $Z_x$ primarily governing 3D structure and $\widehat{Z}_h$ encoding chemical identity.

*Table 6.* Latent disentanglement analysis of MolFLAE and MolAlign3D, where MACCS similarity measures 2D semantic similarity and Shape Tanimoto measures 3D geometric similarity.

| Model | Preserving $Z_x$ | | Preserving $\widehat{Z}_h$ | |
|---|---|---|---|---|
| | MACCS Sim (↓) | Shape Sim (↑) | MACCS Sim (↑) | Shape Sim (↓) |
| MolFLAE | 41.0 | 38.4 | 58.2 | 16.8 |
| **MolAlign3D** | **39.8** | **38.8** | **66.8** | **16.7** |

### 4.3.3. LATENT SPACE INTERPOLATION

**Task.** We linearly interpolate between the latent representations of two molecules and decode the resulting trajectories. Smoothness is quantified by the Pearson correlation $r$ between each property and its interpolation position, oriented by endpoint differences. A property is considered to exhibit significant linear variation along the interpolation trajectory if the null hypothesis of zero Pearson correlation is rejected at the 5% significance level, i.e., if $-\log p > -\log(0.05) \approx 1.3$.

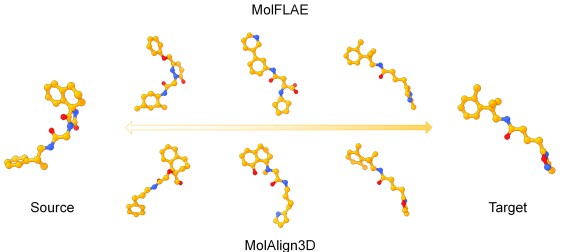

*Figure 4.* Latent space interpolation. MolAlign3D generates valid intermediate structures with smooth property transitions.

**Results.** As shown in Fig. 4 and Table 5, MolAlign3D achieves higher Pearson correlations and stronger significance across structural and physical properties, indicating smoother and more monotonic transitions under linear latent interpolation. Qualitatively, it generates valid intermediates with gradual structural changes, suggesting that semantic anchoring improves the global organization of the fixed-length latent space.

## 5. Conclusion

We presented MolAlign3D, a semantic–generative framework that anchors fixed-dimensional E(3)-equivariant latents with pretrained molecular embeddings. By integrating a geometry-aware encoder, semantic attachment, and a BFN decoder, MolAlign3D constructs a unified latent manifold that is both semantically informative and geometrically precise. Empirically, it substantially improves 3D

*Table 5.* Interpolation performance on latent space. We report Pearson's $r$ and $-\log p$ values for both structural and physical molecular properties across different interpolation numbers ($N$).

| $N$ | Model | Structural Properties | | | | | | | | Physical Properties | | | | | | | |
| | | Similarity Pref. | | sp3frac | | BertzCT | | QED | | Labute ASA | | TPSA | | LogP | | MR | |
| | | $r$ | $-\log p$ | $r$ | $-\log p$ | $r$ | $-\log p$ | $r$ | $-\log p$ | $r$ | $-\log p$ | $r$ | $-\log p$ | $r$ | $-\log p$ | $r$ | $-\log p$ |
| 8 | MolFLAE | 0.9274 | 3.3459 | 0.4031 | 0.7933 | 0.6398 | 1.6865 | 0.5361 | 1.2227 | 0.9073 | 4.4801 | 0.5718 | 1.3775 | 0.5357 | 1.2412 | 0.8240 | 3.3959 |
| | **MolAlign3D** | **0.9412** | **3.5961** | **0.5170** | **1.1319** | **0.7269** | **2.2089** | **0.5465** | **1.2848** | **0.9212** | **4.6813** | **0.6203** | **1.5469** | **0.5930** | **1.4550** | **0.8530** | **3.6297** |
| 10 | MolFLAE | 0.9198 | 4.1487 | 0.3902 | 0.9541 | 0.6273 | 2.0519 | 0.5192 | 1.4478 | 0.9095 | 5.7123 | 0.5556 | 1.6308 | 0.5330 | 1.5155 | 0.8312 | 4.2895 |
| | **MolAlign3D** | **0.9355** | **4.4937** | **0.4878** | **1.3240** | **0.7092** | **2.6978** | **0.5325** | **1.5274** | **0.9229** | **5.9884** | **0.6022** | **1.8703** | **0.5736** | **1.7169** | **0.8573** | **4.6193** |
| 12 | MolFLAE | 0.9118 | 4.8428 | 0.3756 | 1.0548 | 0.6151 | 2.4317 | 0.4975 | 1.6626 | 0.8996 | 6.8405 | 0.5454 | 1.8883 | 0.5287 | 1.7690 | 0.8231 | 5.0541 |
| | **MolAlign3D** | **0.9312** | **5.3641** | **0.4856** | **1.5125** | **0.6993** | **3.1734** | **0.5191** | **1.7758** | **0.9160** | **7.1535** | **0.5976** | **2.1608** | **0.5731** | **2.0357** | **0.8475** | **5.4833** |

reconstruction over MolFLAE while maintaining competitive unconditional generation performance.

Moreover, semantic anchoring improves latent-space behavior, yielding smoother interpolations, more stable atom-count edits, and clearer separation between geometric and chemical information in hybrid latent swaps. The same anchored latent also supports strong molecular representations, achieving competitive downstream property prediction without sacrificing manipulability.

While MolAlign3D demonstrates strong performance across diverse tasks, several limitations remain. First, our editing experiments are currently zero-shot and unconditioned, and extending the framework to pocket-aware molecular design is an important direction. Moreover, while semantic anchoring improves latent disentanglement, the gains are still modest, and achieving more explicit separation between geometry and semantics remains an open direction.

In summary, MolAlign3D shows that anchoring pretrained semantic representations within fixed-dimensional E(3)-equivariant latents provides a powerful and manipulable substrate for 3D molecular modeling. We believe this paradigm offers a general pathway toward unifying molecular representation learning and controllable generative modeling.

## Conflict of Interest Disclosure

The authors declare that they have no financial conflicts of interest related to this work. This research was conducted purely for academic purposes, and no commercial entity was involved in the funding, development, or evaluation of the proposed method.

## Acknowledgements

This work was supported by the Innovative Drug Research and Development-National Science and Technology Major Project (No. 2025ZD1803103), the Beijing Frontier Research Center for Biological Structure, and the Beijing Natural Science Foundation (Qiyan Program, Grant No. 25QY0038). We also thank the anonymous reviewers for their constructive feedback.

## Impact Statement

This paper presents work whose goal is to advance the field of Machine Learning. There are many potential societal consequences of our work, none of which we feel must be specifically highlighted here.

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

## A. E(3)-Equivariance of the Virtual-Node Encoder

**Encoder architecture.** The encoder backbone $\boldsymbol{\Phi}_\theta^{\mathrm{E}(3)}$ is implemented as a stack of $L$ E(3)-equivariant message-passing layers operating jointly on atomic nodes and virtual nodes. At each layer $\ell$, every node $i$ (atomic or virtual) is associated with a coordinate $\mathbf{x}_i^\ell \in \mathbb{R}^3$ and a feature embedding $\mathbf{h}_i^\ell \in \mathbb{R}^{D_{\mathcal{Z}}}$. Message passing alternates between feature updates and coordinate updates.

Feature updates take the form

$$\mathbf{h}_i^{\ell+1} = \mathbf{h}_i^\ell + \sum_{j \in \mathcal{N}(i)} f_h\big(\mathbf{h}_i^\ell, \, \mathbf{h}_j^\ell, \, \|\mathbf{x}_i^\ell - \mathbf{x}_j^\ell\|^2\big) \,, \tag{24}$$

where $\mathcal{N}(i)$ denotes the neighborhood index set of node $i$ in the message-passing graph. $f_h$ depends only on node features and squared pairwise distances, which are invariant under E(3) transformations.

Coordinate updates are defined by

$$\mathbf{x}_i^{\ell+1} = \mathbf{x}_i^\ell + m_i \sum_{j \in \mathcal{N}(i)} (\mathbf{x}_i^\ell - \mathbf{x}_j^\ell) \, f_x\big(\mathbf{h}_i^\ell, \, \mathbf{h}_j^\ell, \, \|\mathbf{x}_i^\ell - \mathbf{x}_j^\ell\|^2\big) \,, \tag{25}$$

where $f_x$ is a scalar-valued function and $m_i \in \{0, 1\}$ is a fixed update mask.

When the architecture is used as the *encoder* backbone in MolAlign3D, the mask enables updates for virtual nodes while keeping the coordinates of real molecular atoms fixed. Conversely, when used in the decoder, the mask is inverted so that atomic coordinates are updated while virtual nodes remain fixed.

**Theorem A.1** (E(3)-Equivariance of the Encoder). *Let $\boldsymbol{\Phi}_\theta^{\mathrm{E}(3)}$ denote the encoder backbone defined by Eqs. (24)–(25). For any rigid transformation $(R, t) \in \mathrm{E}(3)$ applied to the input coordinates,*

$$\mathbf{x} \; \mapsto \; R\mathbf{x} + t,$$

*the encoder output coordinates transform equivariantly, while the corresponding feature embeddings remain invariant.*

*Proof.* Squared distances satisfy

$$\|R\mathbf{x}_i^\ell + t - (R\mathbf{x}_j^\ell + t)\|^2 = \|\mathbf{x}_i^\ell - \mathbf{x}_j^\ell\|^2,$$

and hence the inputs to $f_h$ and $f_x$ are unchanged. Therefore, the feature update in Eq. (24) is invariant.

For coordinate updates, relative displacements transform as

$$\mathbf{x}_i^\ell - \mathbf{x}_j^\ell \; \mapsto \; R(\mathbf{x}_i^\ell - \mathbf{x}_j^\ell),$$

which implies

$$\mathbf{x}_i^{\ell+1} \mapsto R\mathbf{x}_i^\ell + t + m_i \sum_{j \in \mathcal{N}(i)} R(\mathbf{x}_i^\ell - \mathbf{x}_j^\ell) \, f_x(\cdot)$$

$$= R\left(\mathbf{x}_i^\ell + m_i \sum_{j \in \mathcal{N}(i)} (\mathbf{x}_i^\ell - \mathbf{x}_j^\ell) \, f_x(\cdot)\right) + t = R\mathbf{x}_i^{\ell+1} + t.$$

Thus, one layer preserves E(3)-equivariance. By induction over $L$ layers, the full encoder backbone is E(3)-equivariant. $\qquad\square$

**Proposition A.2** (Transformation of Latent Variables). *Before encoding, the input molecule is centered by subtracting its atomic centroid,*

$$\mathbf{X}_{\mathcal{M}} \leftarrow \mathbf{X}_{\mathcal{M}} - \frac{1}{N_{\mathcal{M}}} \sum_{i=1}^{N_{\mathcal{M}}} \mathbf{x}^{(i)}, \tag{26}$$

*which removes the global translational degree of freedom.*

*The virtual-node coordinates are initialized as $\mathbf{X}_{\mathcal{Z}}^{(0)} = \mathbf{0}$, while the virtual-node feature embeddings $\mathbf{H}_{\mathcal{Z}}^{(0)}$ are learnable parameters independent of the input molecule.*

*Let* $(\mathbf{Z}_x, \mathbf{Z}_h)$ *denote the virtual-node outputs of the encoder, as defined in Eq.* (3). *Since* $Z_x$ *is obtained from* $\mu_x$ *via isotropic Gaussian noise, the sampling process preserves E(3)-equivariance in distribution. Then, under any rigid transformation* $(R, t) \in \mathrm{E}(3)$ *applied to the input molecule prior to centering,*

$$\mathbf{Z}_x \mapsto R\mathbf{Z}_x, \qquad \mathbf{Z}_h \mapsto \mathbf{Z}_h. \tag{27}$$

*Proof.* Centering removes the effect of global translations. Since $\mathbf{X}_{\mathcal{Z}}^{(0)} = \mathbf{0}$, rotations induce $\mathbf{X}_{\mathcal{Z}}^{(0)} \mapsto R\mathbf{0} = \mathbf{0}$. By Theorem A.1, each message-passing layer preserves E(3)-equivariance of the coordinates, implying $\mathbf{Z}_x \mapsto R\mathbf{Z}_x$.

The feature embeddings $\mathbf{H}_{\mathcal{Z}}^{(0)}$ are independent of coordinates, and feature updates depend only on invariant quantities. Therefore, $\mathbf{Z}_h$ is invariant under rigid transformations. $\square$

## B. Invariance of Semantic Attachment

**Proposition B.1.** *Let* $\mathbf{Z}_h$ *denote the virtual-node feature embeddings produced by the encoder and let* $\mathbf{u} = \mathfrak{f}_{\mathrm{pre}}(\mathcal{M})$ *be a frozen, E(3)-invariant pretrained molecular embedding. After affine moment matching and per-node concatenation,*

$$\widehat{\mathbf{Z}}_h = \mathrm{Concat}\big(\widetilde{\mathbf{Z}}_h, \ \mathbf{U}_{\mathrm{split}}\big), \tag{28}$$

*the resulting features* $\widehat{\mathbf{Z}}_h$ *remain E(3)-invariant.*

*Proof.* By Proposition A.2, $\mathbf{Z}_h$ is E(3)-invariant. Affine moment matching consists only of feature-wise centering and rescaling and thus preserves invariance. The pretrained embedding $\mathbf{u}$ is invariant by construction. Partitioning $\mathbf{u}$ into $\mathbf{U}_{\mathrm{split}}$ and concatenating it to virtual-node features are coordinate-independent operations. Hence $\widehat{\mathbf{Z}}_h$ remains invariant under any rigid transformation. $\square$

## C. Latent Regularization

We regularize the fixed-dimensional latent manifold from three complementary perspectives: (i) distributional control of the equivariant coordinate branch, (ii) distributional control of the semantic feature branch, and (iii) cross-conformer semantic consistency. Together, these terms stabilize latent geometry while enforcing chemically meaningful structure.

### C.0.1. (A) COORDINATE-BRANCH KL REGULARIZATION

Recall that the encoder produces equivariant coordinate latents $\mathbf{Z}_x \in \mathbb{R}^{N_Z \times 3}$. Following the asymmetric variational parameterization in Equation 3, we impose distributional regularization only on the coordinate branch.

We adopt an isotropic Gaussian prior

$$p(\mathbf{Z}_x) = \mathcal{N}(\mathbf{0}, \mathrm{var}_x \mathbf{I}), \tag{29}$$

with a deliberately large variance $\mathrm{var}_x \gg 1$. The variational posterior factorizes over virtual nodes and coordinates:

$$q(\mathbf{Z}_x) = \mathcal{N}(\boldsymbol{\mu}_x, \boldsymbol{\varsigma}_x^2), \tag{30}$$

where $\boldsymbol{\mu}_x, \boldsymbol{\varsigma}_x^2 \in \mathbb{R}^{N_Z \times 3}$. During training, the predicted variance is clamped elementwise as

$$\boldsymbol{\varsigma}_x^2 \leq \mathbf{1}. \tag{31}$$

The resulting KL divergence admits a closed form:

$$\mathcal{L}_{\mathrm{KL},x} = D_{\mathrm{KL}}\big(\mathcal{N}(\boldsymbol{\mu}_x, \boldsymbol{\varsigma}_x^2) \,\|\, \mathcal{N}(\mathbf{0}, \mathrm{var}_x \mathbf{I})\big), \tag{32}$$

which decomposes elementwise as

$$\mathcal{L}_{\mathrm{KL},x} = \frac{1}{2} \sum_{j=1}^{N_Z} \sum_{d=1}^{3} \left( \frac{\mu_{x,jd}^2 + \varsigma_{x,jd}^2}{\mathrm{var}_x} + \log \frac{\mathrm{var}_x}{\varsigma_{x,jd}^2} - 1 \right). \tag{33}$$

This asymmetric design serves two purposes: (i) clamping $\boldsymbol{\varsigma}_x^2$ prevents excessive stochasticity that would destabilize geometric decoding, while (ii) the high-variance prior induces only a weak quadratic penalty on $\boldsymbol{\mu}_x$, encouraging a broad and uniform occupation of the latent coordinate space. A detailed gradient-based analysis is provided in Appendix D.

## C.0.2. (B) FEATURE-BRANCH REGULARIZATION

In contrast to the coordinate branch, the semantic feature latents $\mathbf{Z}_h \in \mathbb{R}^{N_Z \times D_Z}$ are treated deterministically by the encoder. When decoder-side latent noise is enabled (Equation 10), we apply a lightweight quadratic regularizer to the encoder-produced feature latents to prevent uncontrolled growth in their magnitude.

Specifically, in the presence of additive noise at the decoder input, the encoder can partially counteract the induced perturbation by increasing the scale of its feature outputs, which reduces the relative effect of noise through the decoder and leads to systematic norm inflation of $\mathbf{Z}_h$. This behavior is consistently observed in practice (see Fig. 7), where decoder-side noise causes the mean and variance of feature latents to grow rapidly during training.

To suppress this degeneracy, we penalize the squared magnitude of the encoder-produced feature latents:

$$\mathcal{L}_{\mathrm{reg},h} = \frac{1}{2} \, \mathbb{E}\big[\|\mathbf{Z}_h\|_2^2\big]. \tag{34}$$

In practice, this expectation is implemented as a normalized batch average:

$$\mathcal{L}_{\mathrm{reg},h} = \frac{1}{2} \, \frac{1}{|\mathcal{D}| \, N_Z D_Z} \sum (\mathbf{Z}_h \odot \mathbf{Z}_h), \tag{35}$$

where $|\mathcal{D}|$ denotes the number of molecules in the batch and the sum is taken over all molecules and virtual nodes therein. The empirical effect of this regularization is analyzed in Ablation L of the ablation study.

## C.0.3. (C) MULTI-CONFORMER CONTRASTIVE REGULARIZATION

To enforce semantic consistency across different 3D realizations of the same molecule, we introduce a contrastive objective over multiple conformers. Given two conformers $\mathcal{M}_i^{(1)}$ and $\mathcal{M}_i^{(2)}$ of molecule $i$ within a batch of size $B$, we extract their semantic feature latents $\mathbf{Z}_h^{(1)}, \mathbf{Z}_h^{(2)} \in \mathbb{R}^{N_Z \times D_Z}$ and flatten them as

$$\bar{\mathbf{z}}_h^{(1)}(i), \ \bar{\mathbf{z}}_h^{(2)}(i) \in \mathbb{R}^{N_Z D_Z}. \tag{36}$$

We define dot-product similarity logits

$$\ell_{ij} = \alpha \, \bar{\mathbf{z}}_h^{(1)}(i)^\top \bar{\mathbf{z}}_h^{(2)}(j), \tag{37}$$

where $\alpha = 1/\tau$ is a temperature-scaled factor.

The InfoNCE objective treats $(i, i)$ as a positive pair and all $(i, j)$, $j \neq i$ as negatives:

$$\mathcal{L}_{1 \to 2} = -\frac{1}{B} \sum_{i=1}^{B} \log \frac{\exp(\ell_{ii})}{\sum_{j=1}^{B} \exp(\ell_{ij})}. \tag{38}$$

Defining the induced distribution

$$p(j \mid i) = \frac{\exp(\ell_{ij})}{\sum_{k=1}^{B} \exp(\ell_{ik})}, \tag{39}$$

the per-sample loss corresponds to the negative log-likelihood $-\log p(i \mid i)$. Averaging over the batch yields an equivalent cross-entropy form:

$$\mathcal{L}_{1 \to 2} = \frac{1}{B} \sum_{i=1}^{B} \mathrm{CE}(\mathrm{OneHot}(i), \, p(\cdot \mid i)). \tag{40}$$

In practice, we adopt a symmetric formulation:

$$\mathcal{L}_{\mathrm{contra}} = \frac{1}{2} \left(\mathcal{L}_{1 \to 2} + \mathcal{L}_{2 \to 1}\right), \tag{41}$$

which enforces conformer-invariant semantic representations while preserving structural diversity in the coordinate branch.

## D. Analysis of Coordinate-Branch KL Regularization

We analyze the effect of combining a clamped posterior variance $\varsigma_x^2 \leq \mathbf{1}$ during training with a high-variance prior $p(\mathbf{Z}_x) = \mathcal{N}(\mathbf{0}, \text{var}_x\mathbf{I})$, where $\text{var}_x \gg 1$. Importantly, although the posterior variance is used in the KL objective, no stochastic sampling is performed at test time; instead, the decoder operates directly on the posterior mean $\boldsymbol{\mu}_x$. We therefore focus on how variance clamping influences the distribution of latent means through the KL regularization.

Consider a single scalar component $\mu_{x,jd}$ of the coordinate latent, corresponding to virtual node $j$ and spatial dimension $d$. The KL divergence contribution for this component is

$$\mathcal{L}_{\text{KL},x}^{(j,d)} = \frac{1}{2}\left(\frac{\mu_{x,jd}^2 + \varsigma_{x,jd}^2}{\text{var}_x} + \log\frac{\text{var}_x}{\varsigma_{x,jd}^2} - 1\right). \tag{42}$$

Taking the gradient with respect to the posterior mean yields

$$\frac{\partial\mathcal{L}_{\text{KL},x}^{(j,d)}}{\partial\mu_{x,jd}} = \frac{\mu_{x,jd}}{\text{var}_x}. \tag{43}$$

When $\text{var}_x$ is large, the KL-induced restoring force acting on $\mu_{x,jd}$ is weak and linear. As a result, the KL term only softly penalizes deviations of the latent mean from the origin, in contrast to standard VAEs where $\text{var}_x = 1$ leads to strong contraction of latent means.

Clamping the posterior variance to $\varsigma_{x,jd}^2 \leq 1$ removes the model's ability to reduce the KL loss by inflating uncertainty. Consequently, the encoder must distribute information content primarily through the posterior mean $\boldsymbol{\mu}_x$ rather than through large posterior variance. This encourages the coordinate latent means to spread across the available latent space instead of collapsing toward the origin or concentrating along a small number of high-variance directions.

Formally, under variance clamping and a high-variance prior, the KL contribution associated with the mean term simplifies to

$$\mathcal{L}_{\text{KL},x}(\boldsymbol{\mu}_x) \approx \frac{1}{2\text{var}_x}\|\boldsymbol{\mu}_x\|_2^2, \quad (\text{var}_x \gg 1), \tag{44}$$

which imposes only a mild quadratic regularization on the coordinate latent means. This yields a latent space in which posterior means populate a broad and approximately isotropic region, while remaining compatible with a Gaussian prior for downstream latent diffusion and Bayesian Flow decoding.

This behavior is corroborated by the ablation study in Paragraph L.

## E. Bayesian Update Rules

We provide the explicit Bayesian update rules used to propagate the belief state $\boldsymbol{\Theta}_i = (\boldsymbol{\mu}_i^x, \boldsymbol{\rho}_i^x, \boldsymbol{\theta}_i^v)$ across discrete communication steps in the Bayesian Flow Network. Given the previous belief $\boldsymbol{\Theta}_{i-1}$ and a noisy observation $\mathbf{Y}^{(i)} = (\mathbf{Y}_x^{(i)}, \mathbf{Y}_v^{(i)})$ drawn from the sender distribution, the update $\boldsymbol{\Theta}_i = h(\boldsymbol{\Theta}_{i-1}, \mathbf{Y}^{(i)}, \alpha_i, \alpha_i')$ follows from standard Bayesian inference with conjugate likelihoods.

**Coordinate update.** For atomic coordinates, the receiver maintains a Gaussian belief $\mathcal{N}(\boldsymbol{\mu}_{i-1}^x, (\boldsymbol{\rho}_{i-1}^x)^{-1}\mathbf{I})$, where $\boldsymbol{\rho}_{i-1}^x$ denotes the coordinate precision. Combining this prior with the Gaussian sender likelihood yields a closed-form update of the precision and mean:

$$\boldsymbol{\rho}_i^x = \boldsymbol{\rho}_{i-1}^x + \alpha_i, \qquad \boldsymbol{\mu}_i^x = \frac{\boldsymbol{\rho}_{i-1}^x\boldsymbol{\mu}_{i-1}^x + \alpha_i\mathbf{Y}_x^{(i)}}{\boldsymbol{\rho}_i^x}. \tag{45}$$

**Atom-type update.** For discrete atom types, the receiver maintains a categorical belief $\boldsymbol{\theta}_{i-1}^v \in \Delta^{K-1}$. Upon receiving the Gaussian-embedded noisy signal $\mathbf{Y}_v^{(i)}$ with noise factor $\alpha_i'$, the belief is updated via Bayes' rule as

$$\boldsymbol{\theta}_i^v = h(\boldsymbol{\theta}_{i-1}^v, \mathbf{Y}_v^{(i)}, \alpha_i') := \frac{\exp(\mathbf{Y}_v^{(i)}) \odot \boldsymbol{\theta}_{i-1}^v}{\sum_{k=1}^K \exp(\mathbf{Y}_{v,k}^{(i)})(\boldsymbol{\theta}_{i-1}^v)_k}, \tag{46}$$

where $\odot$ denotes element-wise multiplication and $K$ is the number of atom types.

**Bayesian update distribution and flow.** The deterministic update map $h$ defined above induces a Bayesian update distribution over belief states by marginalizing over the sender randomness. Conditioned on the previous belief $\boldsymbol{\Theta}_{i-1}$, the target molecule $\mathcal{M}$, and the global noise schedule $\boldsymbol{\zeta}$, the update distribution is given by

$$p_U(\boldsymbol{\Theta}_i \mid \boldsymbol{\Theta}_{i-1}, \mathcal{M}, \boldsymbol{\zeta}; \alpha_i) = \mathbb{E}_{\mathbf{Y}^{(i)} \sim p_S} \Big[ \delta\big(\boldsymbol{\Theta}_i - h(\boldsymbol{\Theta}_{i-1}, \mathbf{Y}^{(i)}, \alpha_i)\big) \Big], \tag{47}$$

where $\delta(\cdot)$ denotes the Dirac delta distribution. The expectation over $\mathbf{Y}^{(i)}$ removes the stochasticity of individual messages from the sender and yields a deterministic evolution of beliefs.

Owing to the additive property of precision parameters under conjugate likelihoods, the belief after $i$ communication steps depends only on the accumulated noise schedule. Defining $\beta(t_i) = \sum_{j \leq i} \alpha_j$, the resulting Bayesian flow distribution is given by

$$p_F(\boldsymbol{\Theta}_i \mid \mathcal{M}, \boldsymbol{\zeta}; t_i) = p_U(\boldsymbol{\Theta}_i \mid \boldsymbol{\Theta}_0, \mathcal{M}, \boldsymbol{\zeta}; \beta(t_i)). \tag{48}$$

# F. Molecular Property Evaluation Metrics

**Validity**: Fraction of generated molecules whose SMILES strings can be successfully parsed by RDKit.

**Atom Stability**: Ratio of atoms that satisfy chemically valid bonding configurations.

**Similarity Preference**: Defined as

$$\text{Similarity Preference} = \frac{S_t - S_s}{S_t + S_s},$$

where $S_t$ and $S_s$ denote the Tanimoto similarities to the target and source molecules, respectively, computed using MACCS fingerprints (Durant et al., 2002).

**SA** (Ertl & Schuffenhauer, 2009): Estimates the ease of chemical synthesis of a molecule by combining fragment-based contributions with penalties reflecting molecular size and structural complexity.

**QED** (Bickerton et al., 2012): Measures the overall drug-likeness of a molecule based on multiple physicochemical properties.

**Lipinski** (Lipinski et al., 1997): A set of empirical guidelines that characterize drug-likeness by relating molecular physicochemical properties to oral bioavailability.

**sp3frac**: Fraction of $sp^3$-hybridized carbon atoms relative to the total number of carbon atoms.

**TPSA** (Ertl et al., 2000): Topological polar surface area, reflecting molecular polarity and permeability.

**Labute ASA** (Labute, 2000): Accessible surface area describing the extent of solvent exposure.

**logP** (Wildman & Crippen, 1999): Octanol–water partition coefficient indicating molecular hydrophobicity.

**MR** (Wildman & Crippen, 1999): Molar refractivity related to molecular polarizability and volume.

**BertzCT** (Bertz, 1981): A topological complexity index derived from molecular connectivity and structural features.

**Shape Tanimoto**: Measures the similarity between two molecular shapes as the ratio of their overlapping volume to the union of their volumes, computed after a predefined 3D alignment.

**MCS-IoU**: Measures molecular similarity as the ratio between the size of the maximum common substructure shared by two molecules and the size of their structural union.

# G. Hyperparameters

*Table 7.* Hyperparameters used in MolAlign3D.

| Parameter | Value or description |
|---|---|
| **Backbone** | |
| Number of layers | 9 |
| Hidden dimension $D_f$ | 128 |
| Number of attention heads | 16 |
| $k$ (for KNN cutoff) | 32 |
| Activation | RELU |
| **MolAlign3D Encoder** | |
| Number of virtual nodes $N_Z$ | 8 |
| **Latent & Regularization** | |
| Latent dimension(per node, before anchoring) | 16 |
| Pretrained Representation dimension(per node) | 64 |
| $Z_h$ regularization weight | 0.005 |
| $Z_x$ KL weight | 0.05 |
| Contrastive loss weight | 0.1 |
| $var_x$ | 100.0 |
| KL annealing Steps | 2500(GEOM-DRUG),10000(ZINC-9M) |
| **MolAlign3D Decoder** | |
| Decoder $\beta_1$ | 1.5 |
| Coordinate noise scale $\sigma_1$ | 0.03 |
| Minimum diffusion time $t_{\min}$ | $1 \times 10^{-4}$ |
| Latent noise on $Z_h$ | Enabled, $\tau = 0.1$ |
| Sampling steps | 100 |
| **Latent Diffusion Model** | |
| Minimum diffusion time $t_{\min}$ | $1 \times 10^{-4}$ |
| Sampling steps | 100 |
| **Training and Optimization for MolAlign3D** | |
| Train/Val/Test split(molecule-level) | 290867/584/remaining(GEOM-DRUG), 6921421/996/remaining(ZINC-9M) |
| Batch size | 32 (GEOM-DRUG), 64 (ZINC-9M) |
| Gradient accumulation | 1 (GEOM-DRUG), 2 (ZINC-9M) |
| Maximum epochs | 40(GEOM-DRUG), 15(ZINC-9M) |
| Optimizer | Adam |
| Learning rate | $5 \times 10^{-4}$ |
| $\beta_1/\beta_2$ | 0.95 / 0.999 |
| Weight decay | 0 |
| Learning rate scheduler | ReduceLROnPlateau |
| LR decay factor | 0.4 (GEOM), 0.6 (ZINC-9M) |
| Scheduler patience | 3 (GEOM), 10 (ZINC-9M) |
| Minimum learning rate | $1 \times 10^{-6}$ |
| Gradient clipping | Enabled |

## H. Algorithms for Training MolAlign3D Encoder-Decoder

The training procedure first centers atomic coordinates, then encodes each conformer into virtual-node latents $(\mathbf{Z}_x, \mathbf{Z}_h)$ via the E(3)-equivariant backbone. The feature latents are regularized and anchored with pretrained representations through moment-based alignment and split-and-concatenate. A conformer-level contrastive loss is applied to $\mathbf{Z}_h$, and the final anchored latent $\zeta$ is used to train the Bayesian Flow Network decoder. Algorithm 1 summarizes the end-to-end training loop.

---

**Algorithm 1** Training Algorithm of MolAlign3D

---

**Require:** Minibatch of paired conformers $\{\mathcal{M}_b^{(1)}, \mathcal{M}_b^{(2)}\}_{b=1}^B$; number of virtual nodes $N_Z$; trainable parameters $\Theta$; optimizer Optimizer
**Ensure:** Updated parameters $\Theta$
 1: **while** not converged **do**
 2:     Center atomic positions and concatenate the batch
 3:     $(\mathbf{Z}_x, \mathbf{Z}_h, \mathcal{L}_{\mathrm{KL},x}, \mathcal{L}_{\mathrm{reg},h}) \leftarrow \mathrm{Encode}(\mathbf{X}, \mathbf{V})$
 4:        {Encoder (Eq. 1–3) outputs coordinate latents, feature latents, KL loss, and feature regularization.}
 5:     $\mathcal{L}_{\mathrm{contra}} \leftarrow \mathrm{Contrast}(\mathbf{Z}_h^{(1)}, \mathbf{Z}_h^{(2)})$
 6:        {InfoNCE contrastive loss across conformer pairs (Appendix C).}
 7:     $\widehat{\mathbf{Z}}_h \leftarrow \mathrm{Anchor}(\mathbf{Z}_h, \{\mathcal{M}_b^{(i)}\}_{b,i})$
 8:        {Moment alignment and split-and-concatenate (Eq. 6–8).}
 9:     $\zeta \leftarrow [\mathbf{Z}_x; \widehat{\mathbf{Z}}_h]$
10:     Sample communication time $t \sim \mathcal{U}(0,1)$ and broadcast to batch
11:     $\mathcal{L}_{\mathrm{rec}} \leftarrow \mathrm{RecLoss}(\zeta, t, \mathcal{M})$
12:        {BFN reconstruction loss (Eq. 13–16).}
13:     $\mathcal{L} = \lambda_{\mathrm{rec}}\mathcal{L}_{\mathrm{rec}} + \lambda_{\mathrm{KL}}\mathcal{L}_{\mathrm{KL},x} + \lambda_{\mathrm{reg},h}\mathcal{L}_{\mathrm{reg},h} + \lambda_{\mathrm{contra}}\mathcal{L}_{\mathrm{contra}}$
14:     $\Theta \leftarrow \mathrm{Optimizer}(\mathcal{L}; \Theta)$
15: **end while**
16: **return** $\Theta$

---

## I. MolAlign3D Decoding via Bayesian Flow Networks

Conditioned on the anchored latent representation $\zeta$, **MolAlign3D** decoder employs a Bayesian Flow Network (BFN) to iteratively refine a belief state over continuous atomic coordinates and discrete atom types.

---

**Algorithm 2** MolAlign3D Decoder: sample molecules conditioned on anchored latent

---

1: **Input:** decoder network $\Phi$, anchored latent $\boldsymbol{\zeta}$ (or perturbed $\boldsymbol{\zeta}'$ during training), total steps $T$, number of atoms $N_{\mathcal{M}}$, number of atom types $K$, noise schedules $\beta(\cdot)$, $\beta'(\cdot)$

2: **Output:** Sampled molecule $[\hat{\mathbf{X}}, \hat{\mathbf{V}}]$

3: **Helper:** define $\gamma(t) \leftarrow \dfrac{\beta(t)}{1 - \beta(t)}$

4: **Function** UPDATEPARAMS($\hat{\mathbf{x}}, \hat{\mathbf{v}}, t$)

5:     $\boldsymbol{\mu} \sim \mathcal{N}\big(\gamma(t)\,\hat{\mathbf{x}},\; \gamma(t)(1 - \gamma(t))\mathbf{I}\big)$

6:     Draw $\mathbf{y}^v \sim \mathcal{N}\big(\beta'(t)(K\mathbf{e}_{\hat{\mathbf{v}}} - \mathbf{1}),\; \beta'(t)K\mathbf{I}\big)$

7:     $\boldsymbol{\theta}^v \leftarrow [\mathrm{softmax}((\mathbf{y}^v)^{(d)})]_{d=1}^{N_{\mathcal{M}}}$

8:     **return** $\boldsymbol{\mu}, \boldsymbol{\theta}^v$

9: **end function**

10: Initialize belief parameters: $\boldsymbol{\mu} \leftarrow \mathbf{0},\; \boldsymbol{\rho} \leftarrow \mathbf{1},\; \boldsymbol{\theta}^v \leftarrow \dfrac{1}{K}\mathbf{1}_{N_{\mathcal{M}} \times K}$

11: **for** $i = 1$ **to** $T$ **do**

12:     $t \leftarrow \dfrac{i-1}{T}$

13:     Sample model output (in parameter space):

14:         $\hat{\mathbf{x}}, \hat{\mathbf{v}} \sim p_O\big(\boldsymbol{\mu}, \boldsymbol{\theta}^v, \boldsymbol{\zeta}, t\big)$     // draw from decoder's predictive distribution

15:     Update parameters deterministically in parameter space:

16:         $\boldsymbol{\mu}, \boldsymbol{\theta}^v \leftarrow$ UPDATEPARAMS($\hat{\mathbf{x}}, \hat{\mathbf{v}}, t$)

17: **end for**

18: Final sampling (actual molecular sample):

19:     $\hat{\mathbf{X}}, \hat{\mathbf{V}} \sim p_O\big(\boldsymbol{\mu}, \boldsymbol{\theta}^v, \boldsymbol{\zeta}, 1\big)$

20: **return** $[\hat{\mathbf{X}}, \hat{\mathbf{V}}]$

---

## J. Latent Diffusion Model Training and Sampling

We train a latent diffusion model directly on the anchored latent space $\boldsymbol{\zeta} = [\mathbf{Z}_x; \widehat{\mathbf{Z}}_h]$. The encoder is kept frozen, and the denoising network is optimized to predict clean latent representations from noisy inputs under a predefined schedule.

Algorithm 3 details the training procedure, including latent noising, denoising prediction, and parameter updates.

---

**Algorithm 3** Latent Diffusion Model Training

---

**Require:** Training conformers $\{(\mathbf{x}, \mathbf{h})\}$, frozen encoder $\mathcal{E}$, denoising network $\mathcal{F}_\psi$

**Ensure:** Optimized denoising parameters $\psi$

1: Freeze encoder parameters $\phi$

2: Initialize denoising network $\mathcal{F}_\psi$

3: **while** not converged **do**

4:     $\mathbf{x} \leftarrow \mathbf{x} - \mathrm{mean}_{\mathrm{batch}}(\mathbf{x})$

5:     $(\mathbf{Z}_x, \widehat{\mathbf{Z}}_h) \leftarrow \mathcal{E}(\mathbf{h}, \mathbf{x})$                    {Anchored latent encoding}

6:     $t \sim \mathcal{U}\{1, \ldots, T\},\quad \boldsymbol{\varepsilon}^{\{\cdot\}} \sim \mathcal{N}(\mathbf{0}, \mathbf{I})$

7:     $\mathbf{Z}_t^{\{\cdot\}} \leftarrow \sqrt{\bar{\alpha}_t}\mathbf{Z}_0^{\{\cdot\}} + \sqrt{1 - \bar{\alpha}_t}\boldsymbol{\varepsilon}^{\{\cdot\}}$                    $\{\{\cdot\} \in \{x, h\}\}$

8:     $\widehat{\mathbf{Z}}_0^{\{\cdot\}} \leftarrow \mathcal{F}_\psi(\mathbf{Z}_t^x, \mathbf{Z}_t^h, \tilde{t})$

9:     $\mathcal{L}_{\mathrm{LDM}} \leftarrow \sum_{\{\cdot\}} \frac{1}{N_Z} \sum_{j=1}^{N_Z} \|\widehat{\mathbf{Z}}_0^{\{\cdot\},(j)} - \mathbf{Z}_0^{\{\cdot\},(j)}\|_2^2$

10:     $\psi \leftarrow \mathrm{Optimizer}(\mathcal{L}_{\mathrm{LDM}}; \psi)$

11: **end while**

12: **return** $\psi$

---

After training, new molecular structures are generated by sampling from the learned latent diffusion model. Starting from

Gaussian noise in the latent space, reverse diffusion is applied to recover clean anchored latents, which are subsequently decoded into molecular structures using the BFN decoder.

Algorithm 4 summarizes the reverse diffusion and decoding procedure.

---

**Algorithm 4** Latent Diffusion Model Sampling

---

**Require:** Trained $\mathcal{F}_\psi$, frozen decoder $\mathcal{D}$, diffusion steps $T$
**Ensure:** Generated molecule $\mathcal{M}$
1: $\mathbf{Z}_x^{(T)} \sim \mathcal{N}(\mathbf{0}, \mathbf{I})$
2: $\mathbf{Z}_h^{(T)} \sim \mathcal{N}(\mathbf{0}, \mathbf{I})$
3: **for** $t = T, T-1, \ldots, 1$ **do**
4: $\quad \widehat{\mathbf{Z}}_0^{\{\cdot\}} \leftarrow \mathcal{F}_\psi(\mathbf{Z}_t^x, \mathbf{Z}_t^h, \tilde{t})$
5: $\quad \boldsymbol{\mu}_\psi^{\{\cdot\}} \leftarrow \frac{1}{\sqrt{\alpha_t}}\left(\mathbf{Z}_t^{\{\cdot\}} - \frac{\beta_t}{\sqrt{1-\bar{\alpha}_t}}(\mathbf{Z}_t^{\{\cdot\}} - \sqrt{\bar{\alpha}_t}\widehat{\mathbf{Z}}_0^{\{\cdot\}})\right)$
6: $\quad \mathbf{Z}_{t-1}^{\{\cdot\}} \sim \mathcal{N}(\boldsymbol{\mu}_\psi^{\{\cdot\}}, \beta_t\mathbf{I})$
7: **end for**
8: $\mathcal{M} \leftarrow \mathcal{D}(\mathbf{Z}_x^{(0)}, \mathbf{Z}_h^{(0)})$ {BFN decoder generation}
9: **return** $\mathcal{M}$

---

## K. Hyperparameter Search for Property Prediction

We adopt a hyperparameter search space largely consistent with Uni-Mol (Zhou et al., 2023). The detailed ranges are reported in Table 8.

*Table 8.* Search space of hyperparameters.

| Hyperparameter | Values |
|---|---|
| Learning Rate | [5e-5, 8e-5, 1e-4, 4e-4, 5e-4] |
| Weight Decay | [0.0, 1e-5, 1e-4] |
| Batch Size | [256, 128, 64, 32] |
| Dropout (Task Head) | [0.0, 0.1] |

## L. Ablation Studies

We conduct a series of ablation studies to isolate the effects of key design choices in MolAlign3D.

**(1) Is reparameterization and KL regularization on the coordinate branch necessary?** As shown in Table 9, removing coordinate-branch reparameterization and KL regularization leads to a substantial drop in interpolation consistency across all structural and physical properties. In particular, the Pearson correlation coefficients decrease markedly for geometry-sensitive descriptors such as *sp3frac* and *BertzCT*, indicating that the latent space no longer supports smooth and approximately linear transitions.

These results confirm that reparameterization and KL regularization are essential for preserving the linearity and continuity of the coordinate latent manifold.

We further visualize representative interpolation trajectories in Fig. 5. Without coordinate-branch reparameterization and KL regularization, intermediate structures along the interpolation path often exhibit severe geometric artifacts. In particular, molecules near the midpoint of the interpolation tend to collapse into fragmented structures or display highly distorted conformations, in sharp contrast to the smooth and chemically plausible interpolations produced by the full MolAlign3D model.

*Table 9.* Ablation study on coordinate-branch reparameterization and KL regularization. We report Pearson's correlation coefficients for structural and physical molecular properties under latent-space interpolation with different interpolation numbers ($N$). Removing reparameterization and KL regularization on the coordinate branch results in a pronounced degradation in interpolation consistency.

| $N$ | Model | Structural Properties | | | | Physical Properties | | | |
|---|---|---|---|---|---|---|---|---|---|
| | | Similarity Pref. | sp3frac | BertzCT | QED | Labute ASA | TPSA | LogP | MR |
| 8 | MolFLAE | 0.9274 | 0.4031 | 0.6398 | 0.5361 | 0.9073 | 0.5718 | 0.5357 | 0.8240 |
| | MolAlign3D | **0.9412** | **0.5170** | **0.7269** | **0.5465** | **0.9212** | **0.6203** | **0.5930** | **0.8530** |
| | MolAlign3D w/o rep. & KL ($\mathbf{Z}_x$) | 0.8368 | 0.2785 | 0.4151 | 0.2437 | 0.7451 | 0.4602 | 0.3840 | 0.6002 |
| 10 | MolFLAE | 0.9198 | 0.3902 | 0.6273 | 0.5192 | 0.9095 | 0.5556 | 0.5330 | 0.8312 |
| | MolAlign3D | **0.9355** | **0.4878** | **0.7092** | **0.5325** | **0.9229** | **0.6022** | **0.5736** | **0.8573** |
| | MolAlign3D w/o rep. & KL ($\mathbf{Z}_x$) | 0.8175 | 0.2607 | 0.4152 | 0.2475 | 0.7387 | 0.4370 | 0.3740 | 0.5935 |
| 12 | MolFLAE | 0.9118 | 0.3756 | 0.6151 | 0.4975 | 0.8996 | 0.5454 | 0.5287 | 0.8231 |
| | MolAlign3D | **0.9312** | **0.4856** | **0.6993** | **0.5191** | **0.9160** | **0.5976** | **0.5731** | **0.8475** |
| | MolAlign3D w/o rep. & KL ($\mathbf{Z}_x$) | 0.8043 | 0.2428 | 0.4139 | 0.2270 | 0.7325 | 0.4364 | 0.3788 | 0.5942 |

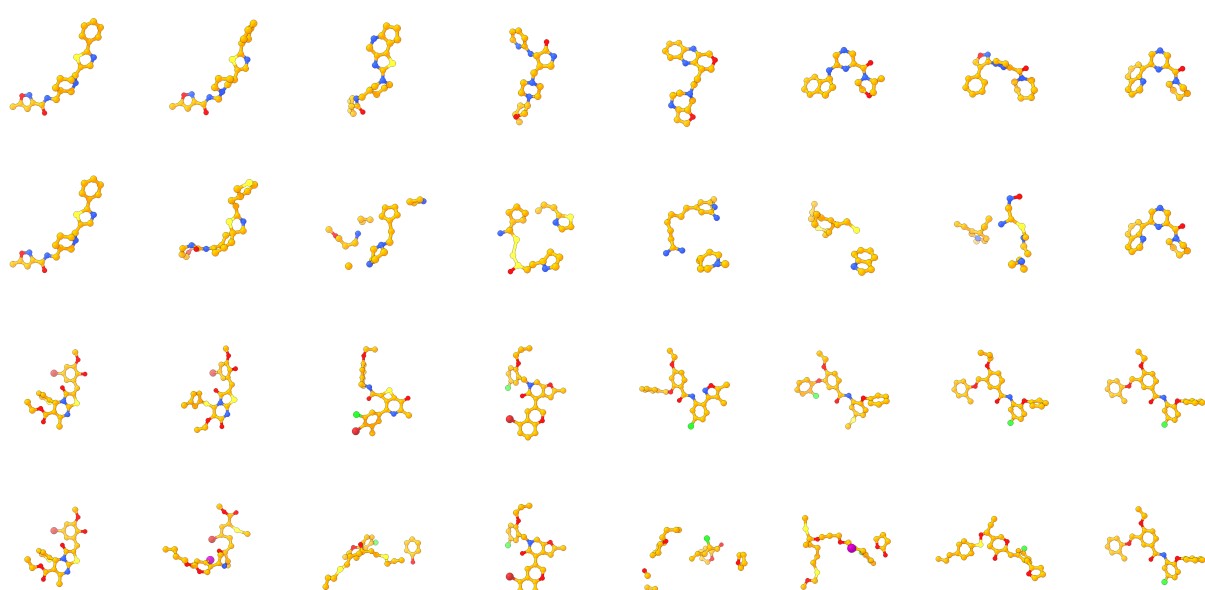

*Figure 5.* Qualitative comparison of latent-space interpolation with and without coordinate-branch reparameterization and KL regularization. The figure contains two representative interpolation trajectories, each spanning two rows. For each trajectory, the source and target molecules are fixed at the two ends, and intermediate structures are generated by linear interpolation in the coordinate latent space. In each pair, the top row corresponds to the full MolAlign3D model with reparameterization and KL regularization on $\mathbf{Z}_x$, producing smooth and structurally consistent transitions. The bottom row shows the corresponding variant without reparameterization and KL regularization, where intermediate geometries often exhibit distorted or discontinuous conformations.

**(2) Is variance clamping on the coordinate branch necessary?** We evaluate the effect of variance clamping on the coordinate latent distribution by removing it while keeping all other components unchanged. Without variance clamping, the coordinate latents exhibit highly concentrated and uneven distributions, indicating poor utilization of the latent space.

As shown in Fig. 6, enabling variance clamping results in a more uniform and well-spread distribution of $\boldsymbol{\mu}_x$, confirming its role in stabilizing the coordinate latent geometry and preventing degenerate concentration.

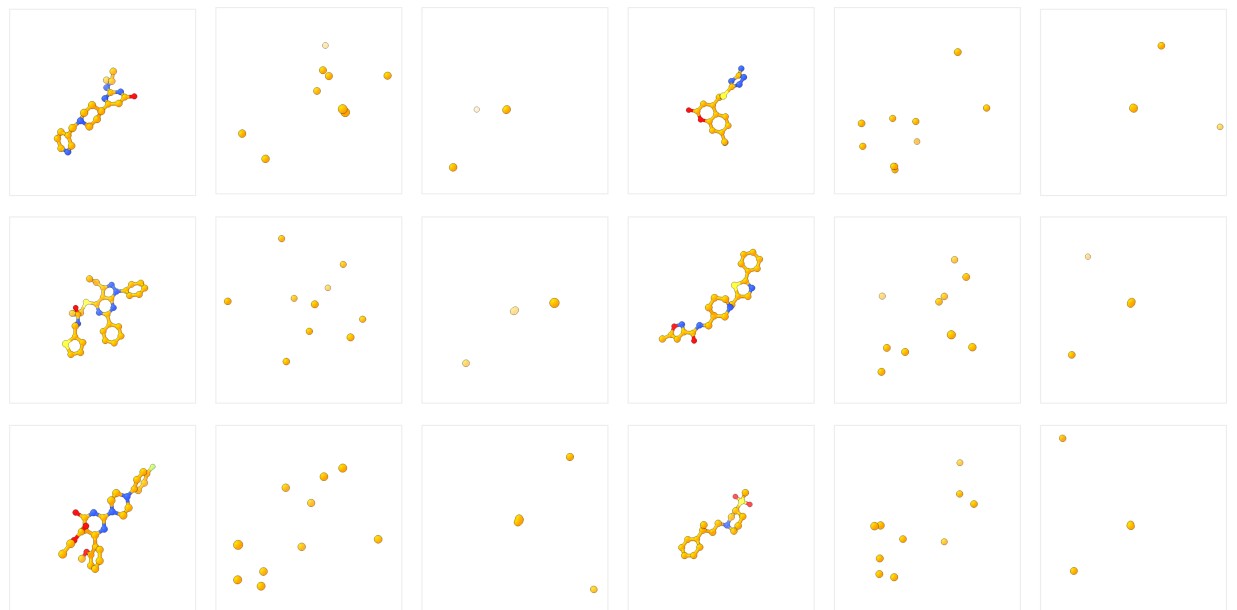

*Figure 6.* Test-time distributions of coordinate latent means $\boldsymbol{\mu}_x$ for representative molecules. Each row shows two molecules, with three panels per molecule: (from left) the reference molecular geometry, the coordinate latent mean $\boldsymbol{\mu}_x$ from a model trained with variance clamping ($\varsigma_x^2 \leq 1$), and that from a model trained without clamping. All latent codes are extracted deterministically at test time, without reparameterized sampling. Variance clamping during training yields more uniformly distributed and approximately isotropic coordinate latent means across conformations, whereas removing the clamp results in concentrated and anisotropic latent distributions.

**(3) Is feature-branch regularization necessary?**    We investigate whether the proposed feature-branch regularization $\mathcal{L}_{\mathrm{reg},h}$ is necessary when decoder-side latent noise is enabled. As discussed in Section C, without explicit control, the encoder can compensate for decoder-side perturbations by increasing the magnitude of its feature outputs, leading to systematic norm inflation of $\mathbf{Z}_h$.

As shown in Fig. 7, the proposed quadratic regularizer effectively stabilizes both the mean and variance of $\mathbf{Z}_h$ throughout training. In contrast, removing this regularization results in rapidly growing feature magnitudes, confirming that the encoder exploits feature scaling to suppress the relative effect of latent noise.

These results demonstrate that feature-branch regularization is critical for preventing degenerative feature scaling and for maintaining a stable and well-behaved semantic latent space.

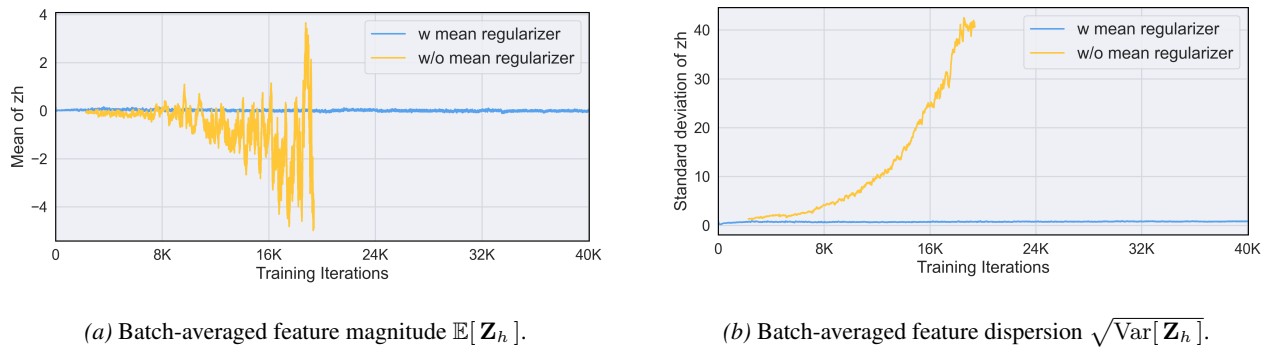

(a) Batch-averaged feature magnitude $\mathbb{E}[\mathbf{Z}_h]$.                    (b) Batch-averaged feature dispersion $\sqrt{\mathrm{Var}[\mathbf{Z}_h]}$.

*Figure 7.* Training dynamics of encoder-produced virtual-node feature latents $\mathbf{Z}_h$ under decoder-side latent noise, with and without the proposed quadratic feature regularizer.

**(4) Why choose split-and-concatenate over broadcasting or attention-based fusion?**    To justify the split-and-concatenate design, we compare it against two natural alternatives: (i) broadcasting the full 512-dimensional Uni-Mol CLS embedding

to every virtual node, and (ii) attention-based fusion between $\mathbf{Z}_h$ and the pretrained semantic embeddings. Table 10 reports reconstruction fidelity on GEOM-Drugs.

*Table 10.* Reconstruction performance of different semantic anchoring strategies.

| Model | MCS-IoU (%, ↑) | Shape Tanimoto (%, ↑) |
|---|---|---|
| MolFLAE | 46.11 | 49.84 |
| Broadcast CLS to all nodes | 58.12 | 54.31 |
| Attention-Based Fusion | 62.39 | 56.25 |
| **MolAlign3D (split-and-concatenate)** | **80.79** | **64.91** |

Broadcasting produces a very high-dimensional latent variable, making latent diffusion harder to model. Attention-based fusion perturbs the pretrained features through learnable interactions, significantly reducing their benefit. Split-and-concatenate keeps per-node dimensionality bounded while structurally allocating semantic information across nodes.

**(5) Is semantic anchoring robust across different pretrained encoders?** We examine whether semantic anchoring remains effective when using a different pretrained source. Replacing Uni-Mol with MolCLR (Wang et al., 2022), Table 11 reports reconstruction and unconditional generation results.

*Table 11.* MolAlign3D with different pretrained encoders.

| Model | Reconstruction | | Unconditional Generation | | | | | |
|---|---|---|---|---|---|---|---|---|
| | MCS-IoU | Shape Tanimoto | Atom Sta. | Valid | QED | SA | Lipinski | LogP |
| MolFLAE | 46.11 | 49.84 | 86.6 | 99.7 | 0.60 | **0.75** | 4.75 | 1.38 |
| MolAlign3D (Uni-Mol) | **80.79** | **64.91** | 87.2 | 99.2 | **0.64** | 0.69 | **4.79** | 2.96 |
| MolAlign3D (MolCLR) | 77.59 | 64.53 | **88.9** | **99.8** | 0.61 | 0.65 | 4.78 | 2.73 |

Regardless of the encoder choice, semantic anchoring brings large improvements over MolFLAE on both reconstruction and generation, suggesting that the benefits come from improved latent space quality rather than any specific pretrained model.

**(6) Is $T = 100$ sufficient for latent diffusion sampling?** We verify whether $T = 100$ sampling steps are sufficient by comparing against $T = 200$ under identical settings. Table 12 shows unconditional generation results.

*Table 12.* Unconditional generation with different numbers of latent diffusion sampling steps.

| Model | Atom Sta. (%) | Valid (%) | QED | SA | Lipinski | LogP |
|---|---|---|---|---|---|---|
| MolAlign3D ($T = 100$) | **87.2** | 99.2 | **0.64** | **0.69** | 4.79 | 2.96 |
| MolAlign3D ($T = 200$) | **87.2** | **99.4** | **0.64** | **0.69** | **4.80** | 2.97 |

This is sufficient in our setting because diffusion is performed in a compressed and well-structured latent space, which significantly reduces the complexity of the generation process. Consequently, increasing the number of steps yields no noticeable improvement across any metric, indicating that 100 steps are sufficient for stable generation and that larger $T$ mainly introduces additional computational cost without clear benefit.

## M. Performance Analysis and Latent Visualization

**Unconditional generation metric fluctuations.** In this paragraph, we explain the small metric fluctuations observed in unconditional generation. Table 2 reports that MolAlign3D achieves strong drug-likeness statistics that closely match the data distribution (e.g., QED, Lipinski, and LogP). Most state-of-the-art models achieve Validity close to 100%, so small fluctuations (e.g., 99.2% vs. MolFLAE's 99.7%) are largely due to stochastic sampling variability and do not indicate structural failures. In addition, the stochastic smoothing applied during training to the feature branch (Eq. 10) enhances robustness and supports zero-shot editing and interpolation, but can also slightly alter the SA metric under unconditional sampling. Overall, MolAlign3D produces molecules that are more consistent with the reference data distribution across multiple complementary metrics.

**Property prediction trade-offs.** MolAlign3D's latent space must simultaneously satisfy E(3)-equivariant reconstruction and semantic alignment, unlike Uni-Mol, which is trained for representation learning without requiring generative reconstruction. Maintaining high-fidelity reconstruction while integrating semantic priors inherently constrains the latent space, so achieving comparable predictive performance already reflects effective integration of semantic information. Notably, MolAlign3D outperforms Uni-Mol on ClinTox, where distinguishing molecules with highly similar structures but different toxicities requires the fine-grained local features captured by the residual virtual-node branch.

**Latent space visualization.** We visualize the latent spaces of MolFLAE, Uni-Mol, and MolAlign3D using t-SNE on the ZINC test set colored by QED. As shown in Fig. 8, MolAlign3D produces a latent distribution with visible property-related structure that closely resembles Uni-Mol, whereas MolFLAE yields a more entangled manifold. This confirms that the proposed design yields a more informative latent space and improves overall performance.

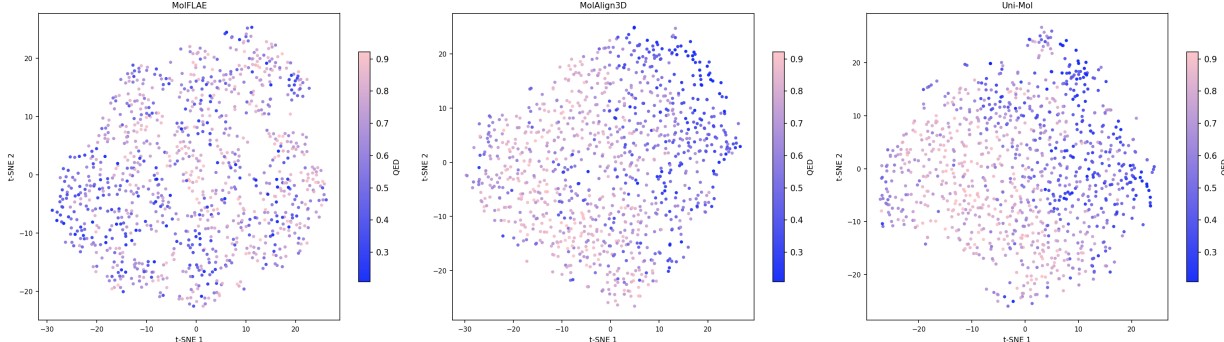

*Figure 8.* t-SNE visualization of latent spaces.

