# OpenReview forum: "MolAlign3D: Enhancing Fixed-Dimensional E(3)-Equivariant Latent Space for High-Fidelity 3D Molecular Reconstruction and Editing"
_ICML.cc/2026/Conference — ICML 2026 regular_

### Official Review · Reviewer_WccA · 2026-03-12

**Soundness:** 3
**Presentation:** 2
**Significance:** 3
**Originality:** 3
**Overall Recommendation:** 3
**Confidence:** 4

**Summary:**

This paper proposes  MolAlign3D, a 3D molecular generation model designed to improve the performance of 3D molecule generation. Existing models inadequately capture semantic information, leading to suboptimal generative performance. MolAlign3D addresses this limitation through two key innovations: (1) anchoring actionable latent features using global semantic embeddings from a pretrained molecular encoder; and (2) integrating geometric and semantic representations into a unified latent manifold.

Experiments evaluate MolAlign3D on 3D molecule reconstruction (e.g., MCS-IoU), unconditional generation (e.g., QED), property prediction (e.g., RMSE), and zero-shot molecular editing. Results show that MolAlign3D generally outperforms models such as UniGEM on datasets including GEOM-Drugs.

**Compliance With Llm Reviewing Policy:**

Affirmed.

**Final Justification:**

The authors' response effectively addressed the main issues I raised; therefore, I increase the Soundness score by 1 point. However, considering the overall contributions and limitations of the paper, I keep the Overall Recommendation unchanged from my original assessment.

**Key Questions For Authors:**

1. It is recommended to expand the “Related Works” section at its end to clearly articulate the core challenges and innovations underlying the design of MolAlign3D.

2. The manuscript should be reviewed to ensure consistent and proper use of variables throughout, with all terms defined upon first appearance.

3. An analysis should be added to explain the reasons behind MolAlign3D’s lower performance relative to MolFLAE and Uni-Mol on specific metrics in Tables 2 and 3.

**Limitations:**

yes

**Strengths And Weaknesses:**

**Strengths**
1. By aligning and fusing actionable latent features with semantic representations through an anchoring strategy, MolAlign3D addresses the issue of semantic insufficiency in 3D molecular generation models. Its generative improvements are validated across multiple tasks, including 3D molecule reconstruction, unconditional generation, property prediction, and zero-shot molecular editing.

**Weaknesses**
1. The “Related Works” section merely surveys existing literature without explicitly articulating the core challenges and innovations that motivated the design of MolAlign3D.

2. Some variables are used inconsistently—for example, the variable $μ$ in Formula 2 is introduced without prior definition.

3. Tables 2 and 3 indicate that MolAlign3D underperforms compared to MolFLAE and Uni-Mol on certain metrics in unconditional generation and property prediction, yet no analysis is provided to explain these results.

---

> ### Author Rebuttal · Authors · 2026-03-31
>
> > ## Weakness1/Question1: About adding a discussion on the core challenges and innovations motivating MolAlign3D in the Related Works section
>
> We acknowledge that the current “Related Works” section does not fully articulate the challenges and innovations motivating MolAlign3D, and we appreciate the reviewer’s suggestion. We will emphasize the core challenges, including but not limited to the lack of a manipulable representation space in existing generative models[1,2], the limited generative ability of standard molecular representation learning methods[3,4], the semantic shallowness of fixed-dimensional latent spaces[5], and the restricted flexibility of prior 3D molecular editing approaches[6,7]. This expansion will clarify the innovations underlying the design of MolAlign3D.
>
> > ## Weakness2/Question2: About consistent variable usage and proper definitions
>
> We thank the reviewer for the careful reading and valuable suggestions regarding the consistency and clarity of variable definitions.
>
> We would like to clarify that the variable $\mu_x$ in Formula 2 is defined here using the “:=” notation, representing the equivariant coordinate output of the encoder for the virtual nodes. We agree, however, that the current presentation is not sufficiently explicit and will provide a clearer explanation of $\mu_x$ in the revised version.
>
> Following the reviewer’s suggestion, we have reviewed the entire manuscript and identified several places where variable definitions or notations can be improved, for example,
>
> 1. The feature dimensionality $D_z$ in Section 3.1 was not explicitly introduced,
> 2. The diffusion variance parameter $\beta_t$ in Formula 21 lacked a clear prior definition,
> 3. In the Appendix, the neighborhood index set $N(i)$ in Formula 24 and the dataset size $|D|$ were not sufficiently explained.
>
> We have now revised the manuscript to ensure consistent notation, explicit variable definitions, and improved readability throughout.
>
> > ## Weakness3/Question3: About deeper analysis on MolAlign3D’s underperformance in unconditional generation and property prediction.
>
> We thank the reviewer for pointing out the need for a deeper analysis of MolAlign3D’s performance in unconditional generation and property prediction. We acknowledge that the original manuscript did not sufficiently explain the slight underperformance on certain metrics.
>
> Regarding unconditional generation (Table 2), most state-of-the-art models achieve Validity close to 100%, so small fluctuations (e.g., 99.7% vs. 99.2%) are largely due to stochastic sampling variability and do not indicate structural failures. In addition, stochastic smoothing applied during training to the feature branch enhances robustness and supports zero-shot editing and interpolation, but can also slightly alter the SA metric under unconditional sampling.
>
> Regarding property prediction (Table 3), MolAlign3D’s latent space must simultaneously satisfy E(3)-equivariant reconstruction and semantic alignment, unlike models such as Uni-Mol, which are trained for representation learning without requiring generative reconstruction. Maintaining high-fidelity reconstruction while integrating semantic priors inherently **constrains the latent space**, so achieving comparable predictive performance already reflects effective integration of semantic information.
>
> We thank the reviewer for this comment and will include a detailed analysis in the revised manuscript.
>
> [1] Hoogeboom et al.,  Equivariant Diffusion for Molecule Generation in 3D, ICML, 2022.
>
> [2] Song et al., Unified Generative Modeling of 3D Molecules with Bayesian Flow Networks, ICLR, 2024.
>
> [3] Zhou et al., Uni-Mol: A Universal 3D Molecular Representation Learning Framework, ICLR, 2023.
>
> [4] Feng et al., Fractional Denoising for 3D Molecular Pre-training, ICML, 2023.
>
> [5] Chen et al., Manipulating 3D Molecules in a Fixed-Dimensional E(3)-Equivariant Latent Space, NeurIPS, 2025.
>
> [6] Guan et al., LinkerNet: Fragment Poses and Linker Co-Design with 3D Equivariant Diffusion, NeurIPS, 2023.
>
> [7] Chen et al., Deep lead optimization enveloped in protein pocket and its application in designing potent and selective ligands targeting LTK protein, Nature Machine Intelligence, 2025.

---

> > ### Author Rebuttal · Reviewer_WccA · 2026-04-03
> >
> > Thank you for the authors' detailed response. I acknowledge the explanations and the commitments to revise for all the issues, and I have decided to maintain my current score.

---

> > > ### Author Response · Authors · 2026-04-04
> > >
> > > Thank you very much for acknowledging that our rebuttal has resolved your main concerns regarding the Related Works, variable consistency, and performance analysis. We appreciate your constructive feedback and will incorporate these improvements into the final manuscript.
> > >
> > > We noticed that your current overall recommendation remains unchanged, primarily based on your assessment of "the overall contributions and limitations of the paper." After carefully reviewing your comments, we were unable to identify specific concerns regarding limited contributions. Could you please clarify what aspects of the contributions or limitations you find insufficient? This would allow us to provide a meaningful explanation or further discussion. We sincerely hope you can give us the opportunity to continue this dialogue.
> > >
> > > Thank you again for your time and thoughtful engagement throughout the review process. Please feel free to share any additional concerns. We are very open to further discussion.

---

### Official Review · Reviewer_PLNB · 2026-03-13

**Soundness:** 2
**Presentation:** 1
**Significance:** 3
**Originality:** 2
**Overall Recommendation:** 4
**Confidence:** 3

**Summary:**

This paper proposes MolAlign3D, a method designed to support both representation learning and generative modeling of 3D molecular structures. Building upon MolFLAE [1], which introduces fixed-length molecular embeddings via virtual nodes, the proposed approach combines molecular structure embeddings with latent representations from a pretrained molecular representation learning model, to capture both structural and semantic information within a unified framework.

Experiments are conducted on both discriminative and generative tasks. On discriminative benchmarks, MolAlign3D outperforms its direct baseline, MolFLAE, and achieves performance comparable to UniMol [2], the pretrained model it builds upon. On generative tasks, including structure reconstruction, editing, and interpolation, the proposed method also demonstrates improvements over MolFLAE.

---

[1] Chen et al., “Manipulating 3D Molecules in a Fixed-Dimensional E(3)-Equivariant Latent Space.” NeurIPS (2025).

[2] Zhou et al., “Uni-Mol: A Universal 3D Molecular Representation Learning Framework.” ICLR (2023).

**Compliance With Llm Reviewing Policy:**

Affirmed.

**Final Justification:**

After checking the authors' rebuttal and the discussions among reviewers, I am satisfied that the main concerns have been sufficiently addressed. No critical issues remain, and the additional experiments and clarifications provided during the discussion period strengthen the paper.

**I therefore raise my score from weak reject to weak accept.**

**Key Questions For Authors:**

1. The interpolation results between two molecular structures appear promising. Could the authors explore whether property-guided interpolation is also feasible within the proposed framework? For instance, generating structures with desired target properties would be highly relevant to practical applications such as drug discovery and inverse materials design.
2. Since the proposed method incorporates Uni-Mol's pretrained representations into the virtual node features, achieving reasonable property prediction performance seems like an expected outcome rather than a distinct contribution. Could the authors discuss what the proposed framework adds beyond what is already provided by the pretrained representations in the discriminative setting?

**Limitations:**

The authors partially address the limitations of their work, but I believe the following points could be further addressed.
* The generative results in Figures 2-4 are not easy to interpret at a glance. Highlighting the key structural differences between generated and reference molecules would make the comparisons significantly more accessible to readers.
* The paper draws inspiration from vision-domain methods but does not sufficiently discuss what challenges or adaptations arise when transferring these ideas to the molecular domain. Including such a discussion would help readers better assess the generality and novelty of the proposed approach.
* The method section describes the pipeline in an incremental manner, making it difficult to follow the overall approach. Providing a concise high-level overview at the beginning of the section would greatly improve readability.

Overall, while the core motivation of this paper is valuable, the limited methodological clarity and the lack of in-depth analysis weaken the contribution in its current form. ***I therefore lean toward a weak reject and encourage the authors to address these issues in the rebuttal.***

**Strengths And Weaknesses:**

Strengths:
* The paper addresses a well-motivated problem at the intersection of 3D molecular generation and representation learning. Extending MolFLAE to incorporate semantic information from pretrained models is a reasonable approach for bridging the gap between these two lines of work.
* The experimental scope covers both discriminative and generative tasks, and the results are reasonable overall. The method achieves discriminative performance comparable to Uni-Mol while demonstrating clear improvements over MolFLAE on the generative side.

Weaknesses:
* The method is presented in a lengthy and incremental manner, making the overall methodology difficult to follow. Figure 1 does not sufficiently clarify the pipeline, and the full algorithm in the Appendix reveals considerable complexity that is not well conveyed in the main manuscript.
* The novelty of the proposed method appears somewhat incremental. Conceptually, MolAlign3D can be viewed as MolFLAE augmented with pretrained molecular embeddings. While this integration is reasonable, the paper does not clearly demonstrate whether the improvements stem from the proposed architectural design or simply from incorporating strong pretrained representations.
* The paper reports experimental results across multiple tasks but lacks in-depth analysis of why the proposed method works.

---

> ### Author Rebuttal · Authors · 2026-03-31
>
> > ## Weakness1/Limitation3: About the Method Presentation
>
> We sincerely apologize for the lack of clarity and thank the reviewer for highlighting this issue. We agree that the method is presented in an overly incremental manner, making the overall approach difficult to follow. In the revision, we will:
>
> 1. add a brief overview at the start of the Methods section to summarize the pipeline;
> 2. redesign Figure 1 to better separate stages and indicate data flow;
> 3. simplify and reorganize the algorithm for clearer presentation in the Appendix.
>
> Thank you again for helping improve the clarity of our paper.
>
> > ## Weakness2: About whether the improvements depend on architectural design or simply on strong pretrained representations
>
> We thank the reviewer for the comment. MolAlign3D’s improvements arise from both architecture and pretrained embeddings. Our split-and-concatenate strategy preserves pure semantic content while keeping latent nodes compact. Ablation studies (see Reviewer 2gMi, Weakness1/Question1&Weakness2/Question2) show that alternative attention-based fusion architecture **underperforms even with strong embeddings**, and that MolAlign3D **with a weaker pretrained encoder (MolCLR[1]) still outperforms it**. These results indicate that the architectural design itself is crucial, as a well-structured latent space allows pretrained embeddings to be effectively leveraged for accurate 3D reconstruction and generation.
>
> > ## Weakness3: About in-depth analysis of why the method works
>
> Thank you for this helpful comment. We apologize for the limited explanation of the mechanism behind MolAlign3D’s performance gains.
>
> The improvements primarily stem from how our design combines synergistic sources of information. The encoder extracts local 3D geometric features, while pretrained molecular embedding provides high-level global semantic context. By fusing these, MolAlign3D organizes the latent space into a more coherent and semantically meaningful structure while preserving fine-grained geometry.
>
> To support this, we conducted an additional analysis by visualizing the latent spaces of MolFLAE, Uni-Mol, and MolAlign3D using t-SNE. We observe that MolAlign3D exhibits clearer property-related patterns and structure, resembling Uni-Mol, whereas MolFLAE shows a more entangled distribution. This suggests that the proposed design yields a more informative latent space and improves overall performance. We will include these visualizations in the camera-ready version to clearly illustrate why the method works.
>
> > ## Question1: About property-guided manipulation
>
> We thank the reviewer for the suggestion. As a preliminary toy experiment, we performed gradient-based optimization in MolAlign3D’s latent space on ESOL[2] for 4 steps to increase the target property. The results are summarized below:
>
> |Step|1|2|3|4|
> |-|-|-|-|-|
> |Predicted Increase|0.0197|0.0393|0.0589|0.0783|
> |Real Increase|0.0234|0.0189|0.0361|0.0754|
>
> These results show that latent-space gradients in MolAlign3D can guide molecules toward higher property values. We appreciate the reviewer for highlighting this promising direction.
>
> > ## Question2：About MolAlign3D’s added value beyond Uni-Mol features in discriminative tasks
>
> Thank you for the comment. While Uni-Mol provides strong global features, MolAlign3D adds **residual, fine-grained structural features** via virtual nodes, capturing subtle local variations.
>
> For example, on the Clintox[2] task, which requires the model to distinguish molecules with **similar structures but different toxicities**, MolAlign3D outperforms Uni-Mol. It shows that these local features make the latent space more discriminative and capture structure-property relationships beyond the pretrained embeddings.
>
> > ## Limitation1: About highlighting key structural differences in figures
>
> We sincerely thank the reviewer for this helpful suggestion. To improve interpretability, we will add highlights to emphasize key structural differences. For example, in Figure 3, we will annotate that MolAlign3D accurately preserves the quinoxaline moiety, a privileged scaffold in medicinal chemistry, at the lower end of the molecule during atom-number editing operations. Similar annotations will be added across Figures 2–4 to make structural comparisons more accessible to readers.
>
> > ## Limitation2: About discussing challenges in adapting vision-domain methods to molecular modeling
>
> We thank the reviewer for the suggestion. We agree that the challenges of transferring vision-domain methods to molecular modeling are not fully discussed. We will expand the Related Works section to cover key challenges and adaptations, including but not limited to ensuring molecular E(3)-equivariance and addressing practical engineering difficulties.
>
> [1] Wang et al., Molecular contrastive learning of representations via graph neural networks, Nature Machine Intelligence, 2022.
>
> [2] Wu et al., MoleculeNet:A Benchmark for Molecular Machine Learning, CORR, 2017.

---

> > ### Author Rebuttal · Reviewer_PLNB · 2026-04-03
> >
> > I appreciate the authors' efforts, but I found the information provided on the following points insufficient to make a full assessment:
> >
> > * Property-guided manipulation
> > * MolAlign3D's added value beyond Uni-Mol features in discriminative tasks
> > * Challenges in adapting vision-domain methods to molecular modeling
> >
> > Could the authors provide further details on these aspects? I will finalize my recommendation after reviewing the additional information and the discussions with other reviewers.

---

> > > ### Author Response · Authors · 2026-04-05
> > >
> > > We sincerely thank the reviewer for the careful reading and for highlighting these important points. We are happy to provide details on each aspect.
> > >
> > > > ## Property-guided manipulation.
> > >
> > > For property-guided manipulation, we randomly select a property (ESOL) from MoleculeNet and train a lightweight predictor $f$ on top of MolAlign3D latent representations. During manipulation, we perform gradient ascent in the latent space. Given a latent variable $z$, we update it via $$z\leftarrow z +\eta \nabla_z f(z)$$ and track the predicted improvement at each step. To evaluate actual chemical changes, we decode the updated latent variables into molecules and compute ESOL scores using RDKit. The results show that the predicted values increase consistently, and the real ESOL of decoded molecules also exhibit an overall upward trend, indicating that latent updates lead to meaningful changes.
> > >
> > > We note that our current experiment focuses on gradient-based manipulation. If the reviewer is interested in other forms of property guidance (e.g., analyzing property variation during interpolation), we would be happy to further clarify or extend the evaluation accordingly.
> > >
> > > > ## MolAlign3D's added value beyond Uni-Mol features in discriminative tasks
> > >
> > > In MolAlign3D, the latent representation is constructed as $$\hat{Z}_h = \mathrm{Concat}(\tilde{Z}_h, U\_{split}),$$ where the first part $\tilde{Z}_h$ is learnable during reconstruction training, while the second part $U\_{split}$ is directly derived from pretrained representations and kept fixed.
> > >
> > > During reconstruction, $\tilde{Z}_h$ is optimized to **capture fine-grained structural details, including local geometry and higher-order interactions that are not fully represented by pretrained embeddings**. This instance-specific information complements pretrained features, enhancing the capture of subtle structure–property relationships. As a result, the model achieves improved performance on tasks like Clintox, where molecules with highly similar structures can exhibit different toxicities due to small local variations.
> > >
> > > Moreover, **to ensure that** $\tilde{Z}_h$ **captures information consistent across different conformers of the same molecule**, we introduce a contrastive objective(see Appendix C.0.3) that **encourages representations of the same molecule to be close while pushing apart those of different molecules**. This is crucial for MoleculeNet tasks, where **labels are defined at the SMILES level and multiple conformers share the same property**, ensuring consistent predictions across conformers.
> > >
> > > > ## Challenges in adapting vision-domain methods to molecular modeling
> > >
> > > Adapting ideas from vision to molecular modeling is non-trivial due to several constraints. **First, molecular structures are variable-sized and symmetry-constrained, requiring permutation invariance and E(3)-equivariance**, which standard vision models lack. **Second, molecules couple discrete chemical identities with continuous 3D geometry**, demanding joint modeling of combinatorial constraints (e.g., atom types and bonds) and precise spatial arrangements, unlike the homogeneous signals in vision. **Third, molecular modeling must account for multiple conformations of the same molecule** that differ in 3D geometry but share identical properties.
> > >
> > > **To address the first challenge,** we design MolAlign3D as a semantic–geometric latent framework rather than directly transferring vision architectures. **We build upon a fixed-dimensional, E(3)-equivariant latent space** to handle symmetry-constrained structures, decoupling representation size from graph size while preserving permutation invariance. Meanwhile, molecule-level semantics are injected **only into the invariant feature components** via alignment and concatenation, **preserving equivariance**. This yields a unified latent space where the coordinate branch captures 3D geometry and the feature branch encodes chemically meaningful semantics.
> > >
> > > **To address the hybrid discrete–continuous nature of molecular representations**, we **decouple the latent representation into a coordinate branch and a feature branch**, and **progressively impose distributional constraints over latent coordinates and features** (see Appendix C.0.1 & C.0.2), which regularizes the joint latent manifold and ensures a consistent coupling between chemical identity and 3D geometry.
> > >
> > > **Finally, to enforce consistent encoding of different conformers**, we **generate multiple conformations for each molecule** and, as mentioned above, **leverage the contrastive objective** introduced for conformer-invariant representation learning.
> > >
> > > Together, these designs bridge key gaps between vision and molecular domains, enabling a structured latent space for both representation learning and high-fidelity molecular generation. These challenges also arise broadly in structured 3D domains such as proteins and materials, suggesting that our design principles may generalize beyond small-molecule modeling.

---

### Official Review · Reviewer_DWRF · 2026-03-14

**Soundness:** 3
**Presentation:** 3
**Significance:** 3
**Originality:** 3
**Overall Recommendation:** 4
**Confidence:** 4

**Summary:**

In this work, the authors have proposed a framework that combines molecular reconstruction with property specific editing by building upon the MolFLAE framework. With this approach, they are able to generate molecular embeddings that incorporate the structural as well as semantic knowledge, enabling both reconstruction and property prediction with a unified latent space of fixed dimension.

**Compliance With Llm Reviewing Policy:**

Affirmed.

**Key Questions For Authors:**

- Why is concatenation used for semantic anchoring? Have the authors tried projection or alignment loss for aligning the two embeddings? Also, how does the splitting preserve node-level semantics?
- Have the authors compared MolAlign3D's performance by swapping out Uni-Mol with other pre-trained models?
- What type of diffusion model was used for generating the latent codes? What was the noise schedule used and what sampling strategy was used? Have the authors performed an ablation on the number of sampling timesteps?

**Limitations:**

yes

**Strengths And Weaknesses:**

Strengths:
- The problem statement is well motivated - there are models for generating molecules and for property prediction, but very few have attempted to unify the two tasks. Since the tasks are very much related, merging them might help augment each other by means of knowledge transfer from one task to another.
- The paper is well written and easy to follow.
- The MolAlign3D framework shows significant improvement in reconstruction task compared to both Uni-Mol as well as MolFLAE. For property prediction as well, the results are comparable to Uni-Mol.
- Ablation studies have been shown for structural disentanglement and interpolation in the latent space.

Weaknesses:
- For enriching the structural representation, MolAlign3D utilizes pretrained encoder models. While this makes the framework agnostic of the encoder, the authors haven't shown the results of using other existing representation learning models like GROVER[1], MolCLR[2], or GEM[3].
- The semantic anchoring mechanism is not theoretically well justified.
- Table 1 shows significant improvement in reconstruction by MolAlign3D over MolFLAE, but the reason behind this is not clear. MolAlign3D essentially uses the same encoding strategy as MolFLAE, only augmented with semantic encoding, but it is not apparent how the semantic encoding helps in structural reconstruction. The reconstruction task is not fine-tuned with the augmented embeddings, so there is no apparent knowledge transfer from the property prediction task to the reconstruction task.
- In section 4.1, the authors mentioned that MolAlign3D uses latent diffusion to generate samples of the latent code, but they have not provided details of the diffusion process. The noise schedule, network architecture, and sampling algorithm have not been mentioned, and the sampling timesteps of 100 also seems very low, which might be because of DDIM sampling which has not been clarified.

[1] Rong et al., Self-supervised graph transformer on large-scale molecular data, NeurIPS, 2020.
[2] Wang et al., Molecular contrastive learning of representations via graph neural networks, Nature Machine Intelligence, 2022.
[3] Fang et al., Geometry-enhanced molecular representation learning for property prediction, Nature Machine Intelligence, 2022.

---

> ### Author Rebuttal · Authors · 2026-03-31
>
> > ## Weaknesses1/Question2: About different pretrained encoders
>
> We are grateful for the reviewer’s comment on this point. While the current version uses a single pretrained encoder for clarity, MolAlign3D is agnostic to the choice of semantic encoder and relies only on meaningful molecular representations. Due to space limitations, we apologize for not including full experimental results here. Please refer to our response to Reviewer 2gMi, Weakness2/Question2, where we report experiments using a relatively weaker pretrained model, MolCLR[1]. These results show that semantic anchoring remains effective across different encoders, with consistent improvements over MolFLAE on both reconstruction and generation tasks.
>
> > ## Weaknesses2/Question1: About the theoretical justification of semantic anchoring
>
> We welcome the reviewer’s thoughtful questions and are happy to clarify our design choices.
>
> > ### 1. Why is concatenation used instead of projection?
>
> Our primary goal is to preserve the purity of the pretrained semantic representation during reconstruction training. Introducing learnable projection would jointly optimize the semantic embedding under the reconstruction loss, causing distortion or entanglement with geometric information. In contrast, simple concatenation injects semantic information without modifying it, allowing it to remain stable and serve as a reliable anchor.
>
> Due to space limitations, we again sincerely apologize and refer the reviewer to our response to Reviewer 2gMi, Weakness1/Question1, for supporting experimental results.
>
> > ### 2. Why not use alignment loss for aligning the two embeddings?
>
> We agree that alignment-based approaches are a natural alternative. However, as noted in prior work such as SVG[2], these methods act as auxiliary constraints without changing the core generative objective, and thus cannot prevent semantic–geometric entanglement. In contrast, our method **explicitly allocates part of the latent space to semantic information**, making it directly accessible and leading to a more stable and interpretable representation.
>
> > ### 3. How does the splitting preserve node-level semantics?
>
> The splitting operation preserves node-level semantics by injecting features individually for each node without aggregation or mixing. Since no cross-node interaction occurs, node-level semantics are maintained explicitly.
>
> > ## Weakness3: About knowledge transfer and the reason for reconstruction improvement
>
> We thank the reviewer for raising this point. Contrary to the concern, **we believe that there is indeed knowledge transfer** from the pretrained semantic representation to the reconstruction task. This is because the decoder is trained on the concatenated, semantically enriched latent, which includes both the encoder-derived structural features and the pretrained semantic embedding.
>
> The improvement in reconstruction arises from the complementary nature of the two components, with the encoder capturing fine-grained local geometry and the pretrained embedding encoding higher-level global semantics. Combining them **allows the decoder to leverage both structural details and global context**, yielding more accurate 3D molecular reconstructions. We observe that this effect is particularly pronounced for larger, more complex molecules, where global semantic cues are crucial for preserving scaffolds and overall geometry.
>
> > ## Weakness4/Question3: About the diffusion model design
>
> We thank the reviewer for pointing out the missing details and will clarify them in the revision.
>
> All other components follow UniMoMo[3], including the cosine variance noise schedule and the standard DDPM ancestral sampling strategy (not DDIM).
>
> For the number of sampling steps, we use 100 steps following UniMoMo. This is sufficient in our setting because diffusion is performed in a compressed and well-structured latent space, which significantly reduces the complexity of the generation process.
>
> To verify this, we conducted an additional experiment with 200 sampling steps, but observed no noticeable improvement across any metrics.
>
> |Model| Atom Sta.(%)|Valid (%)|QED|SA| Lipinski|LogP|
> |-|-|-|-|-|-|-|
> |MolAlign3D-100|87.2|99.2|0.64|0.69|4.79|2.96|
> |MolAlign3D-200|87.2|99.4|0.64|0.69|4.80|2.97|
>
> This suggests that 100 steps are sufficient for stable generation in our setting, and increasing the number of steps mainly introduces additional computational cost without clear benefit. Therefore, we do not further explore larger numbers of sampling steps.
>
> [1] Wang et al., Molecular contrastive learning of representations via graph neural networks, Nature Machine Intelligence, 2022.
>
> [2] Shi et al., Latent Diffusion Model without Variational Autoencoder, 2025.
>
> [3] Kong et al., UniMoMo: Unified Generative Modeling of 3D Molecules for De Novo Binder Design, ICML, 2025.

---

> > ### Author Rebuttal · Reviewer_DWRF · 2026-04-04
> >
> > I thank the authors for the clarifications, especially regarding the semantic anchoring and the knowledge transfer aspect. I suggest the authors add a few more encoder choices to the table (Reviewer 2gMi, Weakness2/Question2) to highlight the agnostic nature of the model, and I keep the current score.

---

> > > ### Author Response · Authors · 2026-04-05
> > >
> > > Thank you for the positive feedback. We are grateful for your time and thoughtful evaluation, and we welcome any further discussion.

---

### Official Review · Reviewer_2gMi · 2026-03-15

**Soundness:** 3
**Presentation:** 3
**Significance:** 3
**Originality:** 2
**Overall Recommendation:** 4
**Confidence:** 3

**Summary:**

This paper extends MolFLAE by introducing semantic anchoring to improve the semantic quality of its fixed-dimensional E(3)-equivariant latent space. The main idea is to combine MolFLAE-style latent nodes with a frozen pretrained molecular embedding: the feature latent ($Z_h$) is aligned to the pretrained embedding distribution, the global embedding is split across latent nodes, and the resulting anchored latent is used for reconstruction, editing, and latent diffusion-based generation.  Experiments show that MolAlign3D substantially improves reconstruction fidelity over MolFLAE, while also improving downstream property prediction and several zero-shot manipulation results.

**Compliance With Llm Reviewing Policy:**

Affirmed.

**Key Questions For Authors:**

1. **Why is the split-and-concatenate semantic anchoring design preferred over alternatives such as directly broadcasting the same global embedding to all latent nodes, using a learned projection, or applying attention-based fusion?**
   A clearer discussion of why this design is preferable to these natural alternatives would help strengthen the methodological motivation.
2. **How sensitive is the proposed method to the choice of pretrained molecular encoder used for semantic anchoring?** The current paper appears to instantiate the framework with one strong pretrained encoder. It would strengthen the paper to clarify whether the proposed semantic anchoring strategy is robust across different pretrained semantic sources.

**Limitations:**

yes

**Strengths And Weaknesses:**

### Strengths
1. **Well-motivated problem formulation and clear presentation.**
The paper is generally well written. The motivation, method, and experiments are aligned around a single central idea: improving the semantic quality of MolFLAE-style latent spaces while preserving geometric fidelity and editability.
2. **Comprehensive Experiments**.
This work evaluates reconstruction, unconditional generation, downstream property prediction, atom-number editing, disentanglement, and interpolation. This gives a reasonably complete empirical picture of the proposed latent space.

### Weaknesses
1. **The motivation behind some core design choices is not fully justified.** The central semantic anchoring step splits a molecule-level pretrained embedding and concatenates the resulting chunks with virtual-node feature latents. This paper does not sufficiently justify why this split-and-concatenate strategy is preferable to simpler or more expressive alternatives.
2. **The method and architecture appear somewhat narrow in scope.** The proposed method feels more like a natural enhancement of MolFLAE via pretrained semantic anchoring than a fundamentally new framework. In addition, the paper offers limited exploration of alternative designs for key components, which makes the method appear architecture-specific rather than broadly general.

---

> ### Author Rebuttal · Authors · 2026-03-31
>
> > ## Weakness1/Question1: About the choice of design for semantic anchoring.
>
> We thank the reviewer for raising this important question regarding the design of the semantic anchoring mechanism. We agree that further clarification of the motivation behind the split-and-concatenate strategy would strengthen the paper.
>
> > ### 1. Why split-and-concatenate instead of broadcasting to all latent nodes?
>
> Our design is motivated by empirical findings in recent works[1,2], which show that diffusion in high-dimensional latent spaces is more challenging and often leads to degraded generative quality.
> In our setting, the pretrained Uni-Mol encoder produces a 512-dimensional CLS embedding, while MolFLAE adopts a moderate per-node latent dimensionality (e.g., each virtual node has a 32-dimensional feature vector). If we directly broadcast the full 512-dimensional embedding to each latent node, the resulting latent variable $\hat{Z_h}$ would become very high-dimensional, making latent diffusion harder to model.
> Instead, the split-and-concatenate design keeps the per-node latent dimensionality bounded, avoiding the difficulty of modeling excessively high-dimensional latents. At the same time, it introduces a structured allocation of semantic information across nodes, rather than redundant duplication as in broadcasting.
>
> > ### 2. Why not use learned projection or attention-based fusion?
>
> Another key design goal is to keep the injected semantic information faithful to the pretrained representation. Adding learnable modules (e.g., projection or attention) would entangle and transform these embeddings during reconstruction, and since semantic preservation is not explicitly enforced by the reconstruction objective, this can distort or overwrite the semantic signal, weakening the benefits of leveraging a strong pretrained encoder.
> Empirically, we verify this via reconstruction experiments:
>
> | Model| MCS-IoU (%, ↑) | Shape Tanimoto (%, ↑) |
> |-|-|-|
> | MolFLAE| 46.11| 49.84|
> | MolAlign3D| 80.79| 64.91|
> | Attention-Based Fusion  | 62.39| 56.25|
>
> When pretrained features are perturbed by attention-based fusion, their benefit is significantly reduced. Although the variant improves over MolFLAE, it still falls notably behind our design.
> This result shows that once semantic features become entangled or distorted, their contribution to reconstruction is already noticeably diminished, let alone their effectiveness in more challenging downstream tasks.
>
> We thank the reviewer for raising this question. We will include these additional results in the revision to better support our design choice.
>
> > ## Weakness2/Question2: About generality and pretrained encoder sensitivity
>
> We thank the reviewer for the insightful comments. Regarding the concern that our method may appear narrow in scope, we agree that our method can be seen as an extension of MolFLAE. However, our goal is not to improve a specific architecture, but to explore a broader unification of molecular generation and representation learning.
>
> Specifically, we aim to learn a unified latent space that supports high-quality generation and reconstruction, captures meaningful molecular semantics, and remains fixed-dimensional for efficient manipulation. MolFLAE serves as a natural starting point due to its generative and fixed-dimensional properties, while our focus is on this broader unification perspective rather than improving MolFLAE itself.
>
> Regarding the reviewer’s concern about alternative designs for key components and the sensitivity to the choice of pretrained molecular encoder for semantic anchoring, we acknowledge that the current version instantiates the framework with a single strong encoder for clarity. To further examine this aspect, we conducted additional experiments using a different pretrained model, MolCLR[3], as the semantic anchor.
>
> Reconstruction (top) and generation (bottom) results:
>
> |Model| MCS-IoU (%) | Shape Tanimoto (%) |
> |-|-|-|
> | MolFLAE|46.11|49.84|
> | MolAlign3D|80.79| 64.91|
> | MolAlign3D-MolCLR|77.59|64.53|
>
> |Model| Atom Sta. (%) | Valid (%) | QED| SA| Lipinski | LogP |
> |-|-|-|-|-|-|-|
> | MolFLAE| 86.6|99.7|0.60 |0.75 |4.75| 1.38 |
> | MolAlign3D| 87.2|99.2|0.64 |0.69|4.79| 2.96 |
> | MolAlign3D-MolCLR |88.9| 99.8|0.61|0.65|4.78| 2.73 |
>
> We observe that, regardless of the choice of pretrained encoder, semantic anchoring brings large improvements in reconstruction over MolFLAE, and also yields consistent gains in generation metrics such as Atom Stability and Lipinski. This suggests the benefits come from improved latent space quality rather than any specific pretrained model.
>
> [1] Zheng et al., Diffusion Transformers with Representation Autoencoders, 2025.
>
> [2] Yao et al., Reconstruction vs. Generation: Taming Optimization Dilemma in Latent Diffusion Models, CVPR, 2025.
>
> [3] Wang et al., Molecular contrastive learning of representations via graph neural networks, Nature Machine Intelligence, 2022.

---

> > ### Author Rebuttal · Reviewer_2gMi · 2026-04-05
> >
> > I appreciate the detailed response. The additional explanations and experiments adequately address my main questions. While I still see the method as a good extension of MolFLAE, I keep my score unchanged.

---

> > > ### Author Response · Authors · 2026-04-05
> > >
> > > Thank you for your thoughtful feedback and for recognizing our additional explanations and experiments. We are grateful for your time and consideration, and we welcome any further discussion.

---

### Decision · Program_Chairs · 2026-04-30

**Decision:**

Accept (regular)

**Comment:**

In this submission, the authors proposed an E(3)-Equivariant method, MolAlign3D, for High-Fidelity 3D Molecular Reconstruction and Editing. Although the reviewers had concerns about the method's implementation and its novelty, the authors' rebuttals addressed these concerns effectively. In summary, I lean towards weak acceptance.